**communications** engineering

# Transforming label-efficient decoding of healthcare wearables with self-supervised learning and "embedded" medical domain expertise

Xiao Gu [1] ✉, Zhangdaihong Liu [1,2] ✉, Jinpei Han [3], Jianing Qiu [4], Wenfei Fang [1], Lei Lu[5], Lei Clifton[1,6], Yuan-Ting Zhang[7] & David A. Clifton[1,2]

Healthcare wearables are transforming health monitoring, generating vast and complex data in everyday free-living environments. While supervised deep learning has enabled tremendous advances in interpreting such data, it remains heavily dependent on large labeled datasets, which are often difficult and expensive to obtain in clinical practice. Self-supervised contrastive learning (SSCL) provides a promising alternative by learning from unlabeled data, but conventional SSCL frequently overlooks important physiological similarities by treating all non-identical instances as unrelated, which can result in suboptimal representations. In this study, we revisit the enduring value of domain knowledge "embedded" in traditional domain feature engineering pipelines and demonstrate how it can be used to guide SSCL. We introduce a framework that integrates clinically meaningful features—such as heart rate variability from electrocardiograms (ECGs)—into the contrastive learning process. These features guide the formation of more relevant positive pairs through nearest-neighbor matching and promote global structure through clustering-based prototype representations. Evaluated across diverse wearable technologies, our method achieves comparable performance with only 10% labeled data, compared to conventional SSCL approaches with full annotations for fine-tuning. This work highlights the indispensable and sustainable role of domain expertise in advancing machine learning for real-world healthcare, especially for healthcare wearables.

The advent of wearable sensors has enabled the collection of vast amounts of medical time series data, both within clinical environments and in daily life[1]. These rich, high-dimensional datasets from wearable devices—such as electrocardiogram (ECG) and photoplethysmogram (PPG) signals for detecting cardiovascular abnormalities[2], electroencephalogram (EEG) signals for monitoring sleep[3], and inertial measurement units (IMUs) for tracking physical activity[4]—offer tremendous potential for real-time health monitoring and early detection of diseases. Analytical solutions for these data have evolved from classical machine learning based on medical domain expertise, namely "*old school*" domain feature engineering, to end-to-end supervised learning that eliminates the need for manual feature extraction from raw data.

In this context, a key challenge lies in actually capturing the *semantics* of these wearable waveform signals, which is the underlying meaning or relevant health/medical information they convey. More specifically, semantic information refers to high-level abstractions aligned with human understanding and clinical knowledge, such as identifying a specific cardiac arrhythmia from cardiac signals, distinguishing sleep stages from brain signals, or profiling behavioral patterns from movement signals. This is in contrast to *syntactic* features, which describe low-level signal characteristics

---

[1]Department of Engineering Science, University of Oxford, Oxford, UK. [2]Oxford Suzhou Centre for Advanced Research, University of Oxford, Suzhou, China. [3]Brain and Behaviour Lab, Imperial College London, London, UK. [4]Department of Biomedical Engineering, The Chinese University of Hong Kong, New Territories, Hong Kong. [5]School of Life Course and Population Sciences, King's College London, London, UK. [6]Nuffield Department of Primary Care Health Sciences, University of Oxford, Oxford, UK. [7]Department of Electronic Engineering, The Chinese University of Hong Kong, New Territories, Hong Kong. ✉e-mail: xiao.gu@eng.ox.ac.uk; zhang.liu@oscar.ox.ac.uk

*syntactic* features (e.g., signal general amplitude or sampling frequency) in the raw waveforms.

Recent advancements in deep learning have substantially improved the automatic extraction of semantic information from raw time series data, such as ECG[2,5,6] and EEG[3,7], and sometimes even surpassing human performance[5]. State-of-the-art methods, built on large-scale annotated datasets, have achieved performance levels comparable to, or even surpassing, those of domain experts. However, in healthcare wearables, manually annotating large datasets, such as ECG or EEG signals, is both labor-intensive and costly[1], often requiring substantial medical domain expertise and, in some cases, additional expensive equipment serving as a "gold standard" (extended discussion is available in Supplementary Material Section S5.1). For example, sleep stage annotation on EEG[8] typically requires polysomnography, a comprehensive recording of physiological signals including brain activity (EEG), eye movement, and muscle activity. These recordings must be examined by trained technicians or sleep specialists, making the annotation process time-consuming and resource-intensive. This creates a significant bottleneck in scaling the use of deep learning for healthcare wearable applications when translating the raw high-dimensional waveforms into actionable clinical insights.

One promising solution to address the scarcity of labeled data is *self-supervised learning (SSL)*, which reduces reliance on annotations by learning useful representations from raw, unlabeled data[9]. In healthcare applications, SSL could streamline the analysis of wearable data, enabling faster and more accurate analytics/diagnoses without the need for labor-intensive data labeling (extended discussion is available in Supplementary Material Section S5.2). Among existing SSL techniques, contrastive learning has been particularly successful[10–15] for its ability to learn representations by aligning semantically similar pairs (e.g., positive pairs) and distancing dissimilar ones (e.g., negative pairs). However, *self-supervised contrastive learning (SSCL)* methods usually require identifying positive and negative pairs without direct access to downstream labels. As such, inappropriate or suboptimal pairings can lead to conflicting optimization objectives, which may limit the model's ability to generalize effectively to real-world medical and healthcare tasks. As illustrated in the bottom panel of Fig. 1, conventional SSCL typically treats two augmentations of the same instance as a positive pair, while all other samples in the batch are considered negatives, even if they may be physiologically similar.

Various strategies have been proposed to find/create appropriate pairs in SSCL, including effective data augmentation techniques, such as adding noise or temporal flipping, that create different augmented views of the same instance[16]. Regarding the former, there are several works that aim to generate domain-inspired augmentation strategies to enhance representation learning[17,18]. However, these methods are often sophisticatedly tailored to specific wearable modalities and lack generalizability across different signal types, due to inherent heterogeneity among modalities. The other line of research resorts to sampling strategies that group positive pairs based on shared medical concepts[10,15]. Typical examples of the latter include treating records from the same patient[19] or from nearby time frames as positive pairs[10]. However, these approaches often introduce inductive biases that may conflict with the underlying medical semantics of the data. For instance, two ECG recordings from the same patient may vary significantly due to sudden cardiac events[20], while recordings from different patients might share similar characteristics if both are healthy. As a result, following such an explicit positive pair grouping policy sometimes may fail to align with the semantic knowledge needed for actual downstream healthcare applications.

Stepping back from the success of deep learning, conventional supervised learning methodologies for healthcare wearables have heavily relied on manually crafted features based on domain-specific knowledge, as shown in the top panel of Fig. 1. Data analysts engineer and abstract these features by leveraging domain expertise, often involving domain experts such as clinicians, who examine the data or use "textbook-level" empirical knowledge to interpret the signals. These features, spanning morphological, temporal, spectral, signal quality dimensions, etc., are then used in classical machine learning classifiers, such as support vector machines or tree-based methods for specific applications[21]. More importantly, these features can be readily extracted using well-documented, community-supported open-sourced toolkits, packages, and repositories developed specifically for physiological signal analysis. For example, the WFDB Toolbox from PhysioNet (https://physionet.org/content/wfdb-matlab/0.10.0/) provides utilities for ECG signal processing, such as R-peak detection, interval measurement, and signal quality assessment, following clinical best practices. This ecosystem is further supported by several community-driven initiatives that have developed robust repositories[22–25] dedicated to domain-specific feature

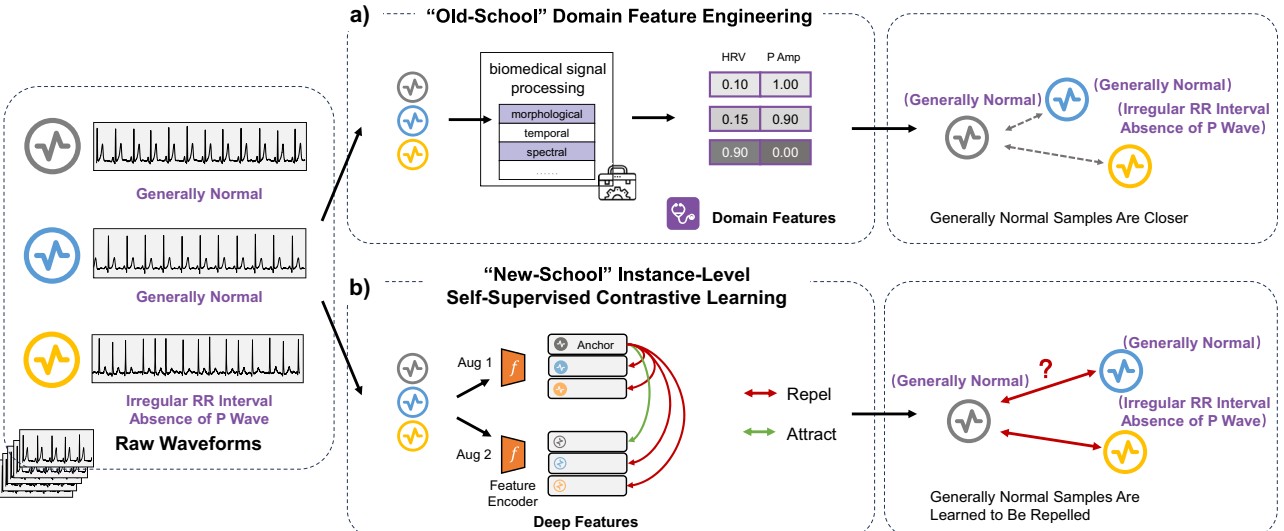

**Fig. 1 | Pipelines of "old-school" domain feature engineering and "new-school" self-supervised contrastive learning (SSCL) for healthcare wearables.** Take the electrocardiogram (ECG) as an example, visualizations illustrate how different ECG patterns are clustered in each pipeline. **a** In the "old-school" pipeline, domain features such as heart rate variability (HRV) and P-wave amplitude (P Amp) are extracted from raw electrocardiogram (ECG) signals using biomedical signal processing tools. They can be used to distinguish abnormal samples against normal samples. **b** In the "new-school" pipeline, two augmented views (Aug 1 and Aug 2) are generated from each ECG input and passed through a shared encoder. The model learns to minimize the distance between the augmented views (positive pair) while maximizing it relative to other samples (negative pairs). ECG electrocardiogram, HRV heart rate variability, P Amp P-wave amplitude.

engineering. They are crafted through extensive reliance on domain expertise, encoding meaningful and clinically relevant concepts specific to the wearable modality[26–28]. Although deep learning approaches have significantly outperformed these traditional techniques in many respects[2,6], the domain knowledge "embedded" underlying such conventional domain feature engineering remains of vital significance, and offers great potential to guide pair selection in self-supervised contrastive learning.

Building on this insight, we propose a domain-knowledge-guided SSCL framework that incorporates and leverages "*old-school*" domain-knowledge-informed features to guide "*new-school*" SSCL. Our framework extends the standard instance-level contrastive learning paradigm (as shown in Fig. 1), in which two augmented views of the same signal are treated as positive pairs and all other signals in the batch are treated as negatives. This conventional setup, however, does not account for the semantic similarities between distinct patients or repeated measurements-potentially treating clinically similar signals as dissimilar during training. To address this, we incorporate domain-specific signal features to guide the formation of more meaningful pairings, thereby improving representation learning in both local and global contexts.

As shown in Fig. 2, our framework introduces two complementary modules:

*Domain-knowledge guided instance-level contrast.* For each training instance, in addition to standard augmentations, we identify semantically similar samples using features extracted via domain-specific signal processing. These nearest neighbors are treated as additional positives during contrastive training, helping to preserve clinically relevant relationships that conventional instance-based SSCL might overlook.

*Domain-knowledge guided prototype-level contrast.* To capture higher-order structure across the whole dataset, we apply unsupervised clustering to the domain-informed features of the training set, prior to training. Each cluster serves as a prototype representing a group of semantically related signals. During training, the network is encouraged to align each sample's representation with its corresponding prototype, thereby reinforcing consistency and robustness at a global level.

Together, these two modules enable our framework to more effectively exploit the latent structure of wearable time series data, without requiring any manual labels. Unlike existing methods that are typically specialized for individual device types/modalities[4,17], our framework is generalizable across a wide range of healthcare wearables that are supported by well-documented, open-source signal processing pipelines for extracting clinically meaningful features. In extensive evaluations across cardiac, neurological, and activity-related wearable datasets, we show that this domain-informed SSCL significantly outperforms existing self-supervised methods-achieving competitive downstream performance with as little as 10% labeled data. We argue that while modern self-supervised techniques have advanced label-efficient learning, domain knowledge "embedded" in conventional domain

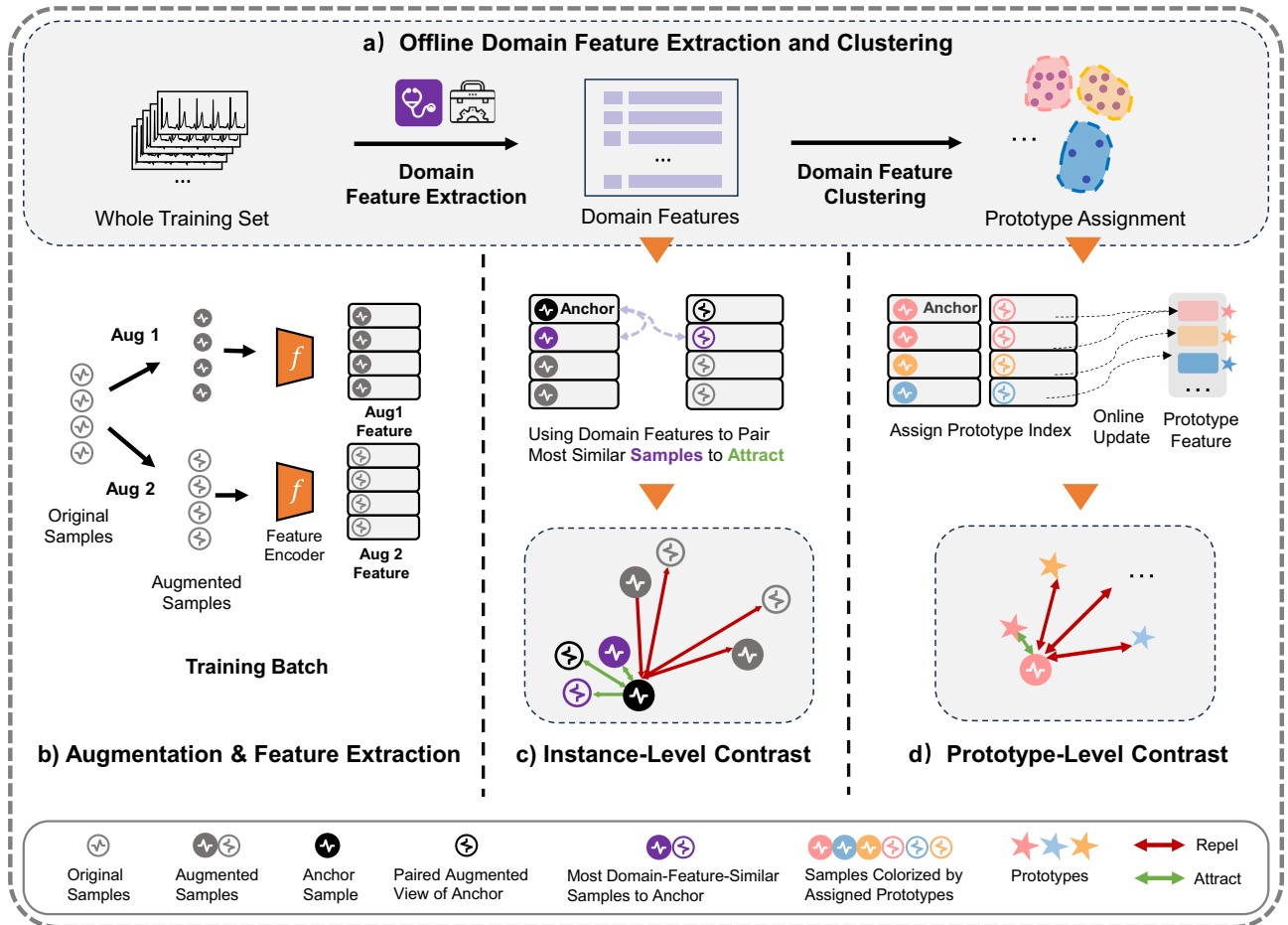

**Fig. 2 | Overview of our domain-knowledge-guided self-supervised contrastive learning (SSCL) framework for decoding wearable time series data. a** Offline domain feature extraction and clustering. Before training, domain features are extracted from raw input signals and clustered using *k*-means to assign a prototype index to each sample. **b** Augmentation and feature extraction. During training, two augmented views (Aug 1 and Aug 2) are generated per sample and passed through a shared encoder to obtain deep features. **c** Instance-level contrast. In addition to treating paired augmented views as a positive pair (marked in black), the nearest neighbor in the domain feature space is selected, and its two augmented views (marked in purple) are included as additional positive. **d** Prototype-level contrast. Each sample is encouraged to align with its domain-informed prototype representation, which is updated online using the encoded features of all assigned samples.

feature engineering remains indispensable for interpreting complex medical waveforms.

# Results

## Overview of the proposed method

**Overall framework.** We developed a domain-knowledge-guided SSL framework to bridge medical domain expertise (embedded in "old-school" domain feature engineering) with advanced deep learning techniques ("new-school" SSCL), for label-efficient wearable data interpretation. The overall framework consists of four stages: (a) offline domain feature extraction and clustering, (b) data augmentation and deep feature extraction, (c) domain-guided instance-level contrast, and (d) domain-guided prototype-level contrast, as illustrated in Fig. 2.

*(a) Offline Domain Feature Extraction and Clustering.* Before training, we extract handcrafted features from all input training samples using domain-specific signal processing tools. These features are computed using open-source domain feature engineering pipelines and reflect established clinical heuristics, such as heart rate variability or waveform amplitude for ECG. The extracted features are then clustered using offline $k$-means, assigning each sample to a prototype group based on domain-informed similarity. This clustering provides a coarse-grained organization of the training set, reflecting broader semantical groupings.

*(b) Augmentation and Deep Feature Extraction.* During training, each input time series undergoes two random augmentations (e.g., jittering, scaling, warping), producing two distinct but related views of the same signal. Both views are passed through a shared encoder network to extract latent representations. This follows the standard SSCL setup, where the model learns by contrasting pairs of representations across the batch.

*(c) Domain-Guided Instance-Level Contrast.* Conventional instance-level SSCL only considers the augmentation from the same signal as positive pairs. In addition to this, we identify additional positive pairs using the similarity of domain features. Specifically, within each batch, based on similarity in the domain feature space, the nearest neighbor of the anchor signal is selected. The two augmented views of this neighbor sample are treated as additional positive, reinforcing the relationship between signals that are semantically similar but not augmented views of the same instance. This process mitigates the risk of treating semantically similar signals as negatives, which is a limitation of conventional instance-level SSCL.

*(d) Domain-Guided Prototype-Level Contrast.* To incorporate global structure, we use the offline cluster assignments from step (a) as a reference for prototype-level contrast. For each cluster, we maintain a prototype representation, which is updated during training using the exponential moving average (EMA) of its members' latent features. During training, each sample is encouraged to align with its assigned prototype in the latent space. This ensures that the model preserves broader structure in the data distribution, helping to stabilize training and improve generalization across diverse physiological conditions.

For technical implementation details, including loss formulations, feature extraction procedures, and training protocols, please refer to the Methods Section.

## Experimental setup

**Datasets.** To assess the feasibility and generalizability of our framework, we conducted experiments across four types of wearable sensing modalities: ECG, EEG, IMU (focusing on 3-Axis Accelerometer), and PPG. All downstream tasks are framed as classification problems, aligning with common healthcare applications associated with each modality, such as physical activity monitoring, cardiac event detection, and sleep stage assessment (Table 1).

The CinC17 dataset contains short ECG recordings collected from portable chest patches, annotated into four categories, including normal sinus rhythm and atrial fibrillation[29]. The CPSC dataset provides 12-lead ECG recordings annotated by cardiologists for nine different cardiac abnormalities, though we used only lead II to simulate wearable scenarios[30]. The MIMIC-III-WDB[31] dataset, consisting of ICU ECG recordings from bedside monitors, was primarily used for self-supervised training due to the absence of segment-level annotations[32]. For EEG, we used the SleepEDF dataset (two channels, Fpz-Cz and Pz-Oz), which includes long-term recordings segmented into 30-s windows and annotated with one of five sleep stages[8]. For IMU, we employed the Capture24 dataset (xyz accelerometer), containing wrist-worn accelerometer recordings segmented into 10-s windows for human activity recognition[24]. Finally, for PPG, we used the Simband dataset[33,34], which comprises 10-s signal segments annotated with cardiac conditions, including normal sinus rhythm, atrial fibrillation, premature atrial/ventricular contractions, and noise. Across all datasets, basic preprocessing was applied, including resampling, band-pass filtering, segmentation, and z-normalization for consistency. All datasets were randomly split into training, validation, and test subsets, in a 6:2:2 ratio, ensuring no subject overlap. They are used to train the models, tune hyperparameters, and evaluate the performance, respectively.

**Domain knowledge-informed features.** Domain-knowledge-informed features were extracted from each modality using established toolboxes and following clinical guidelines. For ECG, 30 features were derived following previous established work[22], including RR intervals and morphological characteristics, which are both key indicators of cardiovascular health, as well as signal quality indices that assess the informativeness and quality of wearable signals[35]. For EEG, 74 features (37 per channel) were computed from both temporal and spectral domains, as per the settings from previous work[23], focusing on spectral power across frequency bands to capture sleep stages[27]. For IMU, 40 features[24] were extracted, including temporal, spectral, and angular measurements, which are essential for activity recognition tasks[36]. For PPG, 102 features were extracted by pyPPG[25], encompassing biomarkers related to peak intervals, component amplitudes, area under the curve, and other clinically relevant metrics. All features were z-normalized to zero mean and unit standard deviation to ensure consistency. A detailed description of these features is provided in the Supplementary Material Section S2.

**Evaluation protocol.** We employed three evaluation protocols to assess the performance and generalizability of the proposed framework: SSL,

**Table 1 | Summary of datasets, modalities, and corresponding real-world applications**

| Dataset | Modality | Downstream task | # Class | # Sample | # Channel | Bandpass | Win (s) | Freq (Hz) |
|---|---|---|---|---|---|---|---|---|
| CinC17[29] | ECG | Cardiovascular Disease Classification | 4 | 8528 | 1 | 0.5–40 | 10 | 125 |
| CPSC[30] | ECG | Rhythm/Morphology Abnormality Identification | 9 | 6877 | 1 | 0.5–40 | 10 | 125 |
| MIMIC-III-WDB[31] | ECG | – | – | 111,619 | 1 | 0.5–40 | 10 | 125 |
| SleepEDF[8] | EEG | Sleep Staging | 5 | 41,509 | 2 | 0.4-30 | 30 | 100 |
| Capture24[24] | IMU | Activity Recognition | 6 | 45,553 | 3 | – | 10 | 100 |
| Simband[33,34] | PPG | Cardiovascular Disease Classification | 4 | 7590 | 1 | 0.5–20 | 10 | 50 |

This table outlines the datasets used in our experiments, the type of wearable modality they represent, and the associated downstream classification tasks. These tasks, based on the expert annotations provided with each dataset, span clinically and behaviorally relevant applications, including cardiac rhythm classification (ECG, PPG), sleep stage detection (EEG), and physical activity recognition (IMU). All tasks are formulated as multi-class classification problems.

semi-supervised learning, and transfer learning. For *self-supervised settings*, we applied K-nearest neighbors (KNN) and linear classifiers (linear probing[9]) to the learned representations for downstream classification tasks across different datasets, separately. These are two simple types of classifiers that do not involve complex feature abstraction, which is well-suited for directly assessing the discriminative power of the learned representations. In the *semi-supervised* setup, we randomly sampled a proportion (5%, 10%, and 20%) of the full training datasets (CinC17 and CPSC) and fine-tuned the entire model using a supervised loss applied only to the available labeled data. This protocol assessed the model's performance when only a limited amount of labeled data is available. For *transfer learning*, the model was pretrained on the larger MIMIC-III-WDB dataset and fine-tuned on fully annotated datasets (CinC17 and CPSC) to evaluate its ability to generalize across different datasets and domains. Across all experiments, performance was measured using the class-average F1 score, AUROC, Precision, Recall, and Accuracy. Measuring the class-average performance of these metrics can indicate the average performance of the model across all classes.

To highlight the superiority of our proposed framework, we compared it against a comprehensive suite of existing SSL methods. These include general-purpose contrastive learning frameworks such as SimCLR[9], BYOL[37], MoCo[38], NNCLR[39], TS[40], SwAV[41], and AMCL[42], all adapted for time series data. We also evaluated domain-specific baselines, including CLOCS[10], originally developed for ECG-based SSL, and TFC[15] and SoftIns[43], both designed for general time series modalities. Additionally, RNC[15] was employed to rank time series representations based on domain features, providing a comparative approach for evaluating how domain knowledge can be incorporated into SSL. For fair comparison, without explicitly mention, all methods were implemented using a ResNet18[44] (1d version) backbone to derive deep features.

Furthermore, we incorporated an additional group of comparisons within the broader *self-supervised learning* landscape. Recent advances in general-purpose time series foundation models[45] have demonstrated the potential to handle diverse temporal signals through large-scale pretraining on heterogeneous datasets[46,47]. To benchmark our framework against this paradigm, we evaluated the representation quality extracted from two representative encoder-decoder-based foundation models: MOMENT[47] and Chronos[46]. These models provide pretrained encoders capable of generating embeddings across a wide range of time series modalities. Implementation details and adaptation procedures are described in the Supplementary Material Section S3.4.

## Overall self-supervised learning performance

**Quantitative analysis of comparison against self-supervised learning methods.** To benchmark the performance of our framework across various healthcare scenarios, we evaluated feature discriminativeness derived from SSL using the class-average F1 score on the test sets (Table 2). Simple classifiers, including a linear classifier and a KNN classifier ($n = 10$), were applied to directly assess the discriminative power of the learned features, as these classifiers do not involve further complex feature abstraction. This allows us to evaluate the quality of the features themselves, rather than introducing additional layers of abstraction that could obscure their inherent effectiveness. We compared our method against a series of SSCL methods, domain feature-based models (Domain Feat.), and fully supervised learning models (Fully Sup.) with randomly initialized backbones, all of which shared the same architecture as our method.

Our method demonstrates superior performance compared to other SSL methods and even outperforms fully supervised models in certain cases, particularly with IMU data, where it achieves a class-average F1 score of 0.526 compared to 0.484. This improvement is likely attributable to our domain knowledge-guided approach, which helps mitigate the effects of inter-subject variability, especially in physical activity data, where individual movement patterns can differ significantly[24]. Further qualitative analysis is presented in Fig. 5. These results suggest that integrating domain knowledge, even in an SSL context, helps reduce biases and enhances the overall robustness of the model.

In addition to class-average F1, we evaluated the best-performing (in terms of class-average F1 score) models on additional metrics, including precision, recall, area under the receiver operating characteristic curve

**Table 2 | Comparison results based on self-supervised learning**

| Methods | KNN | | | | Linear Probing | | | |
|---|---|---|---|---|---|---|---|---|
| | ECG | EEG | IMU | PPG | ECG | EEG | IMU | PPG |
| Fully Sup. | – | – | – | – | 0.641 ± 0.02 | 0.658 ± 0.01 | 0.484 ± 0.05 | 0.520 ± 0.04 |
| Domain Feat. | 0.489 ± 0.05 | 0.392 ± 0.02 | 0.470 ± 0.05 | 0.373 ± 0.04 | 0.509 ± 0.04 | 0.548 ± 0.04 | 0.422 ± 0.06 | 0.377 ± 0.05 |
| SimCLR[9] | 0.427 ± 0.03 | 0.574 ± 0.02 | 0.393 ± 0.04 | 0.395 ± 0.10 | 0.532 ± 0.07 | 0.617 ± 0.03 | 0.433 ± 0.03 | 0.414 ± 0.05 |
| BYOL[37] | 0.380 ± 0.05 | 0.445 ± 0.09 | 0.418 ± 0.10 | 0.243 ± 0.12 | 0.428 ± 0.04 | 0.431 ± 0.10 | 0.449 ± 0.09 | 0.326 ± 0.08 |
| MoCo[38] | 0.450 ± 0.02 | 0.568 ± 0.04 | 0.408 ± 0.04 | 0.368 ± 0.05 | 0.524 ± 0.05 | 0.594 ± 0.08 | 0.425 ± 0.01 | 0.365 ± 0.04 |
| NNCLR[39] | 0.439 ± 0.04 | 0.511 ± 0.04 | 0.383 ± 0.06 | 0.394 ± 0.07 | 0.517 ± 0.05 | 0.555 ± 0.03 | 0.387 ± 0.04 | 0.355 ± 0.06 |
| TS[40] | 0.474 ± 0.05 | 0.580 ± 0.10 | 0.449 ± 0.03 | <u>0.395 ± 0.04</u> | 0.549 ± 0.10 | 0.627 ± 0.03 | 0.501 ± 0.06 | <u>0.452 ± 0.03</u> |
| SwAV[41] | 0.452 ± 0.10 | 0.561 ± 0.05 | 0.438 ± 0.04 | 0.343 ± 0.02 | 0.510 ± 0.08 | 0.638 ± 0.10 | 0.463 ± 0.07 | 0.406 ± 0.03 |
| AMCL[42] | 0.402 ± 0.06 | 0.570 ± 0.07 | 0.413 ± 0.02 | 0.301 ± 0.05 | 0.503 ± 0.04 | 0.626 ± 0.05 | 0.447 ± 0.05 | 0.419 ± 0.07 |
| CLOCS[10] | 0.488 ± 0.01 | <u>0.581 ± 0.06</u> | 0.451 ± 0.05 | 0.376 ± 0.05 | 0.537 ± 0.03 | <u>0.642 ± 0.05</u> | 0.483 ± 0.05 | 0.451 ± 0.06 |
| TFC[15] | 0.405 ± 0.03 | 0.543 ± 0.03 | 0.267 ± 0.04 | 0.383 ± 0.04 | 0.411 ± 0.02 | 0.571 ± 0.07 | 0.307 ± 0.01 | 0.401 ± 0.04 |
| SoftIns[43] | 0.481 ± 0.04 | 0.539 ± 0.03 | 0.420 ± 0.05 | 0.337 ± 0.05 | 0.557 ± 0.10 | 0.602 ± 0.03 | 0.498 ± 0.08 | 0.382 ± 0.06 |
| RNC[49] | <u>0.478 ± 0.04</u> | 0.504 ± 0.08 | <u>0.465 ± 0.01</u> | 0.342 ± 0.02 | <u>0.550 ± 0.03</u> | 0.560 ± 0.02 | 0.517 ± 0.05 | 0.424 ± 0.05 |
| MOMENT[47] | 0.443 ± 0.03 | 0.464 ± 0.02 | **0.517 ± 0.02** | 0.366 ± 0.01 | 0.514 ± 0.03 | 0.456 ± 0.01 | <u>0.522 ± 0.02</u> | 0.411 ± 0.03 |
| Chronos[46] | 0.325 ± 0.03 | 0.452 ± 0.04 | 0.445 ± 0.04 | 0.387 ± 0.05 | 0.301 ± 0.08 | 0.406 ± 0.06 | 0.481 ± 0.06 | 0.410 ± 0.10 |
| Ours | **0.509 ± 0.04** | **0.591 ± 0.07** | <u>0.465 ± 0.02</u> | **0.420 ± 0.04** | **0.574 ± 0.03** | **0.643 ± 0.03** | **0.526 ± 0.04** | **0.499 ± 0.02** |

To test the efficacy of the downstream knowledge learned during self-supervised learning, we froze the backbone as a feature extractor and applied K-nearest neighbors (KNN) and a linear classifier for supervised training. We compared the class-average F1 on the test subset based on the model with the best F1 value on the validation subset. Comparison includes a series of SSCL methods, domain feature-based models (Domain Feat.), and fully supervised learning models (Fully Sup.) with randomly initialized backbones. Each method was run with five different random seeds to assess the performance. We reported the mean ± standard deviation of the five results. Bold indicates the best result, and Underline indicates the second-best result.

(AUC), and accuracy (as shown in Fig. 3). Our method consistently outperformed other SSL approaches across almost all these metrics, further demonstrating its generalizability and effectiveness in learning robust representations from wearable health data.

Compared to state-of-the-art SSCL methods, our approach benefits from the incorporation of domain-knowledge-driven features, which enhances the selection of contrastive pairs and ensures that the model captures clinically meaningful patterns. This finding highlights a promising direction for advancing SSL in time series data, particularly in healthcare applications where domain knowledge plays a critical role in improving model performance.

**Quantitative analysis of comparison against time series foundation models.** In terms of general-purpose time series foundation models, both MOMENT and Chronos were pretrained on large, heterogeneous collections of non-medical time series (e.g., industrial, financial, environmental), and thus lack exposure to high-frequency, high-dimensional physiological signals. While MOMENT outperforms Chronos (e.g., ECG: KNN F1 0.435 vs. 0.325; Linear Probing F1 0.514 vs. 0.301), neither matches our domain-guided SSCL on clinical data. Foundation models capture broad temporal patterns but miss the fine-grained, medically relevant distinctions in ECG, EEG, and PPG. The one exception appears in the IMU KNN evaluation, where MOMENT leads with 0.517 compared to our 0.465, suggesting that large, heterogeneous pretraining can sometimes encode general motion dynamics very effectively. These results underscore the distinct nature of healthcare wearable waveforms and show that pretraining on domain-aligned data, potentially combined with medical knowledge-informed contrastive learning,

yields more robust, generalizable representations. Please refer to Supplementary Material S4.1 for additional results.

**Qualitative analysis of the waveform attention map.** We employed Grad-CAM[48] to visually interpret the learned features of different methods by generating attention maps superimposed onto the original waveforms, as shown in Fig. 4. Grad-CAM highlights key regions of the input data that are most influential in model predictions, allowing for a direct assessment of feature importance. We combined a fully-trained classifier with the learned backbone to produce these attention maps.

For the atrial fibrillation case in Fig. 4, our method effectively captures clinically relevant morphological abnormalities, such as the absence of p-waves—a key indicator of atrial fibrillation, demonstrating its ability to focus on diagnostically critical features. Figure 4 further presents a representative sample of ST-segment elevation. Here, our approach consistently assigns higher saliency to waveform subsegments that directly reflect ST elevation, while effectively ignoring irrelevant or noisy sections. This selective attention to clinically meaningful subsegments not only enhances the interpretability of the model's decision-making but also boosts its overall prediction accuracy (Table 2).

**Qualitative analysis of feature distribution.** Furthermore, we included a t-SNE visualization (Fig. 5) comparing domain features, features from end-to-end supervised learning, those learned by SimCLR, and those learned by our method, based on the whole dataset of Capture24. It can be observed from the first column that domain knowledge-based features are evenly distributed between training/validation/testing subjects. In contrast, features learned via end-to-end supervised training (second

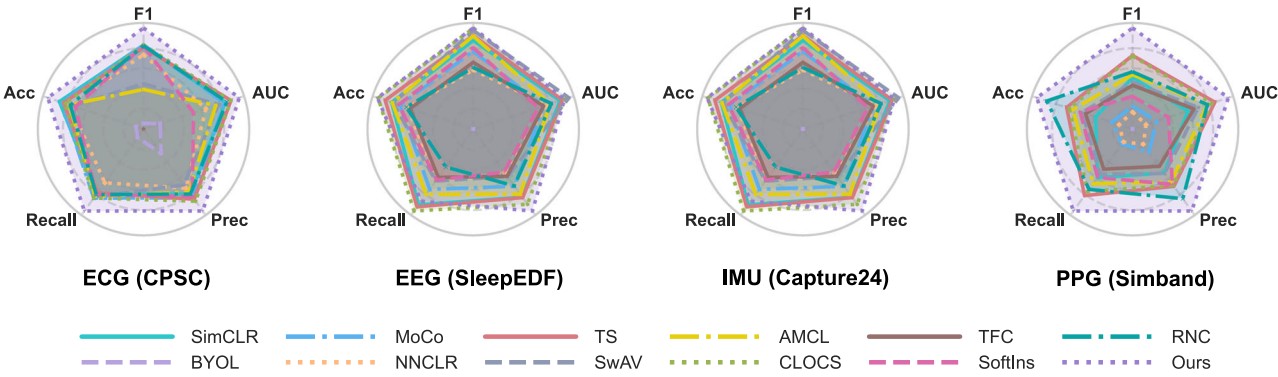

**Fig. 3 | Performance comparison in terms of class-average F1, Accuracy (Acc), Area Under the Receiver Operating Characteristic Curve (AUC), Recall, and Precision (Prec).** We compared the performance on the test subset, based on the model with the best F1 score on the validation subset. The values are normalized by the deviation from the best performance for each respective metric.

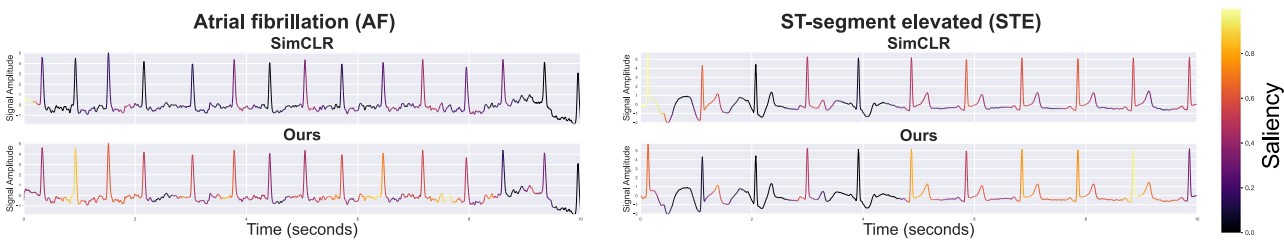

**Fig. 4 | Grad-CAM-based waveform saliency map visualizations for two representative samples.** A fully trained classifier, appended to each self-supervised learning backbone, was used to generate the saliency maps. These maps highlight the most important waveform segments that contribute to the model's predictions. The

saliency values, normalized between 0 and 1 for each sample, allow for improved visual comparison. Our method assigns higher saliency to clinically relevant regions, such as ST-segment elevation, thereby improving both interpretability and prediction accuracy.

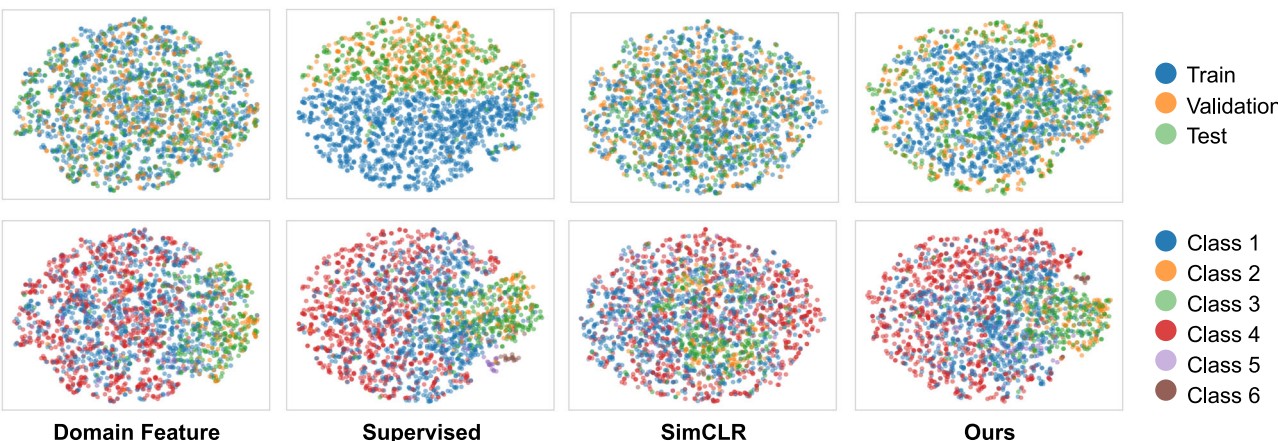

**Fig. 5 | t-SNE visualizations of domain features, features extracted from end-to-end supervised learning, from SimCLR, and from our method, respectively.** Each point corresponds to a sample from the Capture24 dataset. Each column represents the same plots, with points colorized by different manners. In the top panel, each sample points were colored by data split (train/validation/test), whilst in the bottom, shaped by class label.

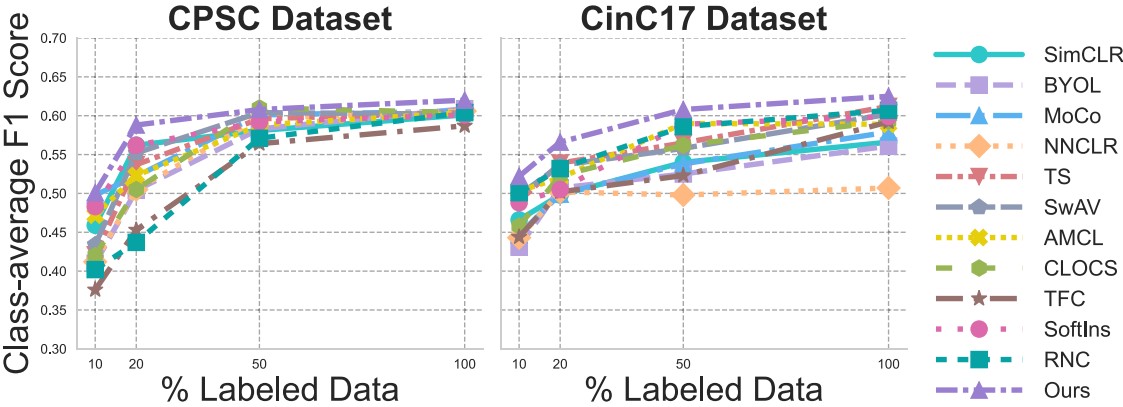

**Fig. 6 | Performance comparison of pretrained models fine-tuned using varying proportions of labeled data under a semi-supervised learning setting on ECG datasets.** Fine-tuning was performed using a supervised loss computed only on the labeled samples. Model performance is reported in terms of class-average F1 score.

column) exhibit clear clustering based on training and validation/test splits. This suggests that the model may have overfitted to subject-specific characteristics, such as individual wearing patterns or sensor noise, which are not directly related to activity-related semantic information.

As shown in the third and fourth columns, SimCLR and Ours, both approaches produce more evenly distributed feature representations across training, validation, and test sets. SimCLR performs instance-level contrast, which implicitly pushes away the representations of the instances from the same subject. This helps to avoid learning those subject-specific characteristics as biases. However, the learned representations are not semantically discriminative enough. As observed from the plot of the second row, third column, points from different classes get more mixed compared to other plots of the same row. Our method goes a step further by using domain features to guide positive pair selection. This helps the model focus on learning semantically meaningful patterns rather than unrelated differences between individuals.

## Semi-supervised with a small proportion of annotations

To further test the efficacy of our method, we conducted experiments in semi-supervised learning settings. We fine-tuned the backbone network using varying fractions of labeled data, specifically with label ratios set to 10%, 20%, 50%, and 100%. Specifically, in a simplistic way, we applied a supervised loss on the labeled subset only to conduct fine-tuning. The results for class-average F1 scores are presented in Fig. 6.

Our method consistently outperforms others, particularly when labeled data is scarce. For instance, with only 10% of labeled data, our

approach demonstrates consistent improvements over competing methods. Additionally, our method is less sensitive to the proportion of labeled data compared to other models, maintaining strong performance even with small amounts of annotated data. This confirms the label efficiency of our proposed framework, as the features learned by SSL already show strong discriminativeness, reducing the reliance on large labeled datasets for effective downstream performance.

## Transfer learning with a large-scale dataset

To demonstrate the transferability of our learned representations across different datasets within the same modality, we utilized the large-scale MIMIC-III-WDB dataset for pretraining. After pretraining, we fine-tuned the entire network on the labeled CPSC and CinC17 datasets. The results, presented in Table 3, show that self-supervised pretraining on the diverse and extensive MIMIC-III-WDB dataset leads to improved performance across all methods. This improvement is likely due to the size and variability of the dataset, as detailed in Table 3.

Among all the methods evaluated, our proposed approach consistently demonstrates superior performance in nearly all cases. This highlights the effectiveness of our method in both small-scale and large-scale data scenarios, suggesting that the SSL pipeline is capable of learning transferable representations that generalize well to other datasets. Our method not only excels in extracting meaningful features for downstream tasks but also proves to be robust across varied healthcare datasets, providing large improvements in transfer learning scenarios.

## Impact of domain knowledge-informed features

**Sensitivity to domain features**. Recognizing the critical role of domain features within our integrated framework, we conducted a comprehensive ablation study on these features in the ECG domain. Specifically, we categorized the features into distinct groups (AF related features, Morphological features, RR interval-related features, Beat similarity-related features) and systematically ablated each group individually. Our results, shown in Fig. 7, illustrate that, traditional logistic regression model trained directly on domain features (in blue) exhibits evident drops in F1 score when any group is removed. This reflects a strong dependency on the presence of specific clinical descriptors. By contrast, our SSCL approach (in green) remains robust, consistently outperforming direct regression even with reduced or non-optimized

feature sets. For example, the weakest feature group under SSCL still achieved 0.545 in F1 score, exceeding the baseline of 0.532 from SimCLR. This indicates that our framework can effectively leverage domain signals-even when they are coarse or weak-without requiring highly discriminative feature design.

This robustness is clinically meaningful. In real-world wearable settings, data quality often varies, and key features—such as clean RR intervals or stable waveform segments—may be degraded by noise, motion artifacts, or incomplete recording. Moreover, the extracted features, while widely used and clinically grounded, may not be tailored to any specific disease. Our framework's ability to tolerate such imperfections and still extract meaningful representations makes it particularly well-suited for practical deployment in diverse and uncontrolled environments.

**Table 3 | Comparison results based on transfer learning on ECG datasets**

| Methods | CPSC | | | | CinC17 | | | |
|---|---|---|---|---|---|---|---|---|
| | F1 | Recall | Prec. | AUC | F1 | Recall | Prec. | AUC |
| Random Init. | 0.585 | 0.599 | 0.605 | 0.905 | 0.536 | 0.599 | 0.547 | 0.841 |
| SimCLR[9] | 0.658 | **0.689** | 0.652 | 0.930 | 0.627 | 0.705 | 0.626 | 0.883 |
| BYOL[37] | 0.658 | 0.672 | 0.654 | 0.925 | 0.630 | 0.712 | 0.616 | 0.880 |
| MoCo[38] | 0.640 | 0.663 | 0.639 | 0.929 | 0.651 | 0.719 | 0.625 | 0.881 |
| NNCLR[39] | 0.640 | 0.667 | 0.640 | 0.926 | 0.644 | 0.717 | 0.620 | 0.885 |
| TS[40] | 0.627 | 0.638 | 0.630 | 0.921 | 0.624 | 0.715 | 0.602 | 0.887 |
| SwAV[41] | 0.654 | 0.670 | 0.650 | 0.929 | 0.642 | 0.716 | 0.628 | 0.889 |
| AMCL[42] | 0.660 | 0.668 | 0.658 | 0.933 | 0.649 | 0.725 | 0.630 | 0.892 |
| CLOCS[10] | 0.647 | 0.671 | 0.637 | 0.927 | 0.652 | 0.740 | 0.619 | 0.887 |
| TFC[15] | 0.640 | 0.661 | 0.650 | 0.922 | 0.648 | 0.717 | 0.614 | 0.893 |
| SoftIns[43] | 0.657 | 0.670 | 0.652 | 0.935 | 0.645 | 0.730 | 0.629 | 0.890 |
| RNC[49] | 0.642 | 0.652 | 0.653 | 0.926 | 0.654 | 0.734 | 0.634 | 0.893 |
| Ours | **0.670** | 0.667 | **0.675** | **0.945** | **0.680** | **0.751** | **0.644** | **0.901** |

Based on the pretrained model with self-supervised learning on the large-scale MIMIC-III-WDB dataset, the results on CPSC and CinC17 are reported, respectively. We compared the class-average F1, AUC, Precision, Recall, and Accuracy on the test subset based on the model with the best F1 value on the validation set. Bold indicates the best result, and Underline indicates the second-best result.

**Fig. 7 | Comparison of feature group importance for direct logistic regression versus our framework.** The heatmap on the left illustrates which combinations of features (AF-Feature, Morphological, RR Interval, and Similarity) were used, where the purple cubes indicate feature utility. The bar plot on the right shows the class-average F1 score for each feature combination, with direct logistic regression represented in blue (hatched) and our self-supervised contrastive learning method in green.

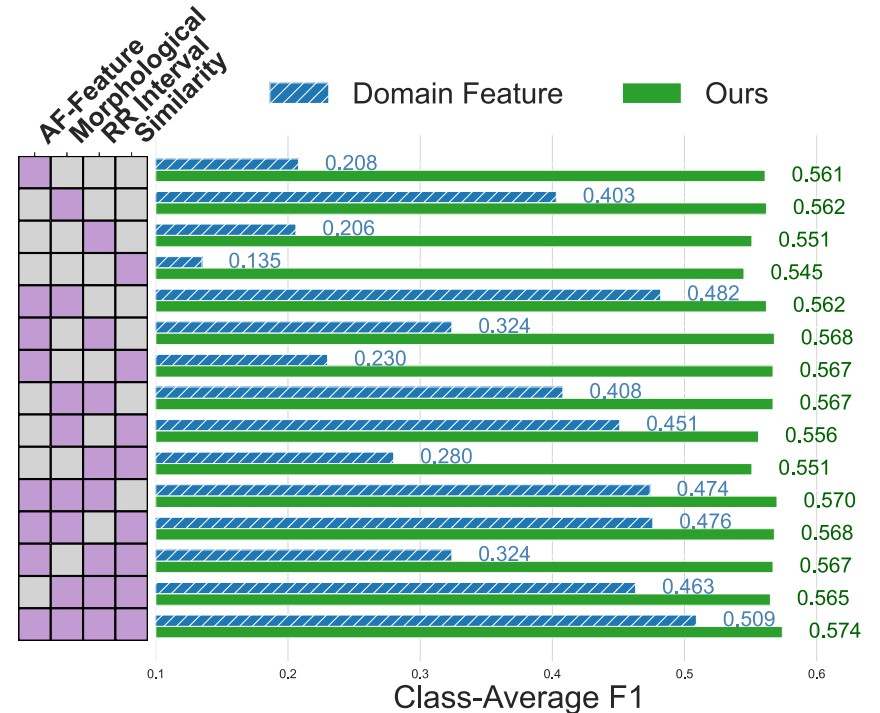

**Comparison of different strategies for using domain features.** We further compared several approaches to incorporate domain knowledge into SSL, with results shown in Table 4. These include (i) Deep Regression, which directly maps deep features to domain features via a multi-layer perceptron module, with the $\mathcal{L}_2$ regression loss as the loss during training; (ii) Ranking, where samples are ranked using Euclidean distances in the domain feature space, and subsequently the ranking is used to guide contrastive learning as in RNC[49]; and (iii) Nearest Neighbour Only, which applies domain-guided instance-level contrast without prototypes. As outlined in Table 4, direct feature regression leads to suboptimal performance. While domain-knowledge-informed features encode semantically meaningful information, they often lack sufficient separability. Relying solely on these features reduces overall performance, reinforcing the importance of integrating them into an SSL framework to enhance both semantic understanding and discriminative capability.

On the other hand, domain-feature-guided ranking achieves performance comparable to nearest neighbour search only, suggesting that domain features offer useful guidance during representation learning. However, both approaches fall short of our full framework, which combines instance-level and prototype-level contrast. This combination leverages both local and global domain-informed relationships, leading to the most robust and discriminative feature representations.

## Model-agnostic nature of our framework

To highlight the flexibility of our framework, we tested its performance across various backbone architectures, including ResNet18, ResNet34, ResNet50, and MSDNN[50], under SSCL conditions. As shown in Table 5, our method consistently demonstrates superior performance across different backbones, further demonstrating its adaptability and robustness. These results confirm that our approach is model-agnostic, capable of achieving strong performance regardless of the underlying architecture, making it versatile for diverse applications in healthcare wearables.

## Discussion

Digital health technologies are becoming increasingly pervasive, leading to the availability of vast amounts of wearable sensing data. This surge in data collection presents a unique opportunity to extract relevant health insights from complex waveforms. However, a critical challenge remains: how to efficiently derive clinically meaningful information from this vast and complex data landscape while requiring minimal manual annotation.

Existing SSCL methods have made progress by reducing reliance on labeled data, but they are limited by their suboptimal formulation of contrastive pairs. These traditional approaches often fail to account for the intricate semantic relationships inherent in health data. In particular, the conventional assumption that samples from different subjects are negative pairs can result in learning representations that fail to capture critical subtleties, such as shared patterns in physiological signals between patients with similar conditions. As a result, the model may fail to capture essential and critical medical insights from the wearable data.

In contrast, the field of biomedical signal processing offers a rich set of bespoke features across temporal, spectral, and morphological aspects. These features have been refined over years of medical expertise and are readily available for use in interpreting time series data. Our work capitalizes on the medical domain expertise "embedded" in these domain features, integrating it into the SSL framework to enhance the learning of semantic representations from wearable health data. This fusion of "*old-school*" domain knowledge with "*new-school*" SSL offers a solution to the limitations of traditional SSCL methods.

The integration of these domain features has proven effective across multiple datasets and wearable devices, including ECG, EEG, IMU, and PPG data. Our method consistently outperformed existing SSCL frameworks, particularly in cases with limited labeled data, which is a common challenge in healthcare. For example, our method demonstrated superior performance in detecting atrial fibrillation in ECG data and distinguishing sleep stages from EEG recordings. This suggests that the use of domain-

**Table 4 | Comparison of domain feature usage strategies on the CPSC (ECG) dataset**

| Methods | F1 |
|---|---|
| Deep Regression | 0.510 |
| Domain Feature Ranking | 0.550 |
| Nearest Neighbour Search Only | 0.550 |
| Ours | 0.574 |

We evaluate four different approaches for incorporating domain features into self-supervised learning. *Deep Regression*: A multilayer perceptron is added to regress deep features to domain features. *Ranking (RNC[49])*: Samples are ranked using Euclidean distance in the domain feature space. *Nearest Neighbour Only*: Our proposed instance-level contrast is based on domain-feature-guided nearest neighbors. *Ours (Full)*: Combines both domain-guided instance- and prototype-level contrast.

**Table 5 | Comparison results of different backbones (ResNet18, ResNet34, ResNet50, and MSDNN) on CPSC (ECG) dataset**

| Methods | ResNet18 | ResNet34 | ResNet50 | MSDNN |
|---|---|---|---|---|
| SimCLR[9] | 0.532 | 0.554 | 0.574 | 0.464 |
| BYOL[37] | 0.428 | 0.400 | 0.527 | 0.328 |
| MoCo[38] | 0.524 | 0.486 | 0.568 | 0.493 |
| NNCLR[39] | 0.517 | 0.515 | 0.555 | 0.424 |
| TS[40] | 0.549 | 0.559 | 0.562 | 0.505 |
| SwAV[41] | 0.510 | 0.520 | 0.518 | 0.480 |
| AMCL[42] | 0.503 | 0.537 | 0.563 | 0.501 |
| CLOCS[10] | 0.537 | 0.557 | 0.560 | 0.533 |
| TFC[15] | 0.411 | 0.412 | 0.550 | 0.373 |
| SoftIns[43] | 0.557 | 0.532 | 0.565 | 0.508 |
| RNC[49] | 0.550 | 0.542 | 0.563 | 0.547 |
| Ours | **0.574** | **0.580** | **0.581** | **0.555** |

We compared the class-average-average F1 on the test subset, based on the model with the best F1 on the validation subset. Bold indicates the best result.

informed features significantly improves the model's ability to generalize across diverse health conditions.

The success of our domain-knowledge-enhanced SSCL approach has important clinical implications. As wearable sensors become more ubiquitous, especially in remote patient monitoring and telemedicine, the ability to accurately interpret these complex signals with minimal annotation becomes crucial. Furthermore, the improvement in model robustness and performance, particularly in low-label or noisy environments, can support healthcare providers in offering more timely and precise interventions based on wearable data. For instance, better detection of arrhythmias in ECG signals or identifying early-stage sleep disorders through EEG data could translate into more proactive and personalized healthcare interventions.

Moreover, the flexibility of our approach allows it to generalize across various healthcare applications. Beyond wearable data, the integration of other data modalities—such as clinical notes, electronic health records (EHRs), and imaging data—could provide richer contextual information, enhancing both diagnostic accuracy and clinical decision-making. This broader context would enable the development of more comprehensive, multi-modal health monitoring systems capable of offering deeper insights into patient health.

## Limitations and future work

While this work presents a promising framework, several limitations should be acknowledged. First, the method's reliance on domain-specific features,

such as RR intervals in ECG data or spectral power in EEG, means that its success is closely tied to the quality of these features. In scenarios where domain-specific features are less well-defined or unavailable, the model's performance may suffer. Future research could focus on developing more generalizable features that do not depend on predefined medical characteristics. An alternative would be to integrate other data modalities-clinical notes, structured tabular health records, and imaging data-which provide richer contextual information.

Additionally, our method has been primarily validated in offline settings. Its real-time performance and applicability on resource-constrained devices, such as those commonly used in wearables or edge computing platforms, remain untested. Future work should explore the deployment of this framework in real-time scenarios, ensuring its ability to handle streaming data efficiently without sacrificing accuracy.

Another limitation lies in the impact of noisy data. Wearable devices are prone to noise from environmental factors or poor sensor placement, which could degrade the quality of domain-specific features. Although our preprocessing addressed basic noise removal, further research is needed to explore the model's robustness to noise in more dynamic, real-world settings.

## Methods
### Problem definition
In this study, we focus on wearable time series data collected from different healthcare modalities. Considering an unlabeled dataset denoted as $D$, consisting of wearable time series data $\{x_i \in \mathbb{R}^{c \times t}\}_{i=1}^{N}$, where $c$, $t$, and $N$ represent the number of signal channels, time steps, and samples, respectively. Our objective is to develop a feature encoder $f : x \to z \in \mathbb{R}^{\dim(z)}$, capable of extracting semantically meaningful representations for downstream healthcare-related tasks. In the present work, these tasks include ECG/PPG-based cardiovascular disease classification, EEG-based sleep staging, and IMU-based activity recognition.

**Basic self-supervised contrastive learning loss**. Most SSCL methods, including the one we build upon, employ a variant of the InfoNCE loss[9]. The goal of contrastive learning is to bring the representations of positive pairs closer together while pushing apart the representations of negative pairs. The general form of the InfoNCE loss used at the instance level is defined as:

$$\mathcal{L}(x_i) = -\log \frac{\exp\left(z_i^\top z_p / \tau\right)}{\exp\left(z_i^\top z_p / \tau\right) + \sum_{n \in \mathcal{N}(i)} \exp(z_i^\top z_n / \tau)}, \quad (1)$$

where $i$, $p$ denote the index of the anchor and positive samples, respectively; $\mathcal{N}(i)$ represents the negative set of sample $x_i$ in a mini-batch, and $\tau$ is the temperature that controls the sharpness of similarity. Out of simplicity and without loss of generality, the latent representation $z$ is processed by another non-linear projector and $l_2$ normalized prior to the similarity calculation in the following, by default.

Typically, in the absence of actual ground truth, $i$ and $p$ are defined as augmented views of the same instance[9]. Consequently, the loss function Eq. (1) encourages the model to learn representations that make the augmented views similar in the latent space while separating them from the representations of other samples in a mini-batch.

### Proposed methodology
**Domain knowledge-based features**. To leverage the guidance of domain knowledge to improve our learning framework, for each $x_i$, we extract a domain knowledge-based feature $d_i$. These features are selected based on their relevance to the corresponding modality-specific domain knowledge and have been identified as important features based on domain-specific expertise[26,27].

The extracted features are precomputed offline using established toolboxes and clinical guidelines, ensuring that they do not impose

additional computational costs during training. Once extracted, these features are z-normalized to ensure consistency across different datasets and modalities. Further details about the feature extraction process and the specific feature types used for each modality can be found in the Supplementary Material Sections S1 and S2.

**Domain knowledge guided nearest neighbour instance-level contrast**. Conventional SSCL methods often face challenges in healthcare applications due to sampling biases. Specifically, instance-based contrastive learning typically assumes that all non-positive samples (i.e., negative samples) are dissimilar, which may lead to the erroneous separation of samples that actually share similar clinical or physiological characteristics. This issue is particularly acute in healthcare data, where different individuals with the same condition might present similar patterns, such as patients with the same type of arrhythmia in ECG data. Misclassifying such semantically similar data as dissimilar could hinder the model's ability to learn clinically meaningful representations.

To overcome this limitation, we perform nearest neighbor search in the domain feature space. By leveraging domain knowledge encoded in domain-specific features (e.g., morphological, spectral, and temporal characteristics), we can identify samples with higher semantic similarity to be treated as additional positive pairs. For each anchor sample $x_i$, the nearest neighbors in a mini-batch are identified based on the Euclidean distance in the domain feature space, as follows:

$$S(i) = \arg\min_{j \in \mathcal{N}(i)} \| d_j - d_i \|_2, \quad (2)$$

where $\mathcal{N}(i)$, $d_i$ represents the set of negative samples and the domain feature vector of sample $i$, respectively. By selecting the most similar samples based on domain features, we ensure that semantically meaningful pairs are identified. In practice, since our method builds upon augmentation-based SSCL approaches to generate positive pairs, and considering that the augmented views of a sample share the same domain feature vector $d$, we select the two most similar samples in the domain feature space to form the $S(i)$. Further empirical analysis of increasing the number of neighbors is available in Supplementary Material Section S4.3.

Building on this, we modify the original instance-level discrimination loss as below,

$$\mathcal{L}_{con}^{ins}(x_i) = -\frac{1}{|\mathcal{P}(i)|} \sum_{p^* \in \mathcal{P}(i)} \log \frac{\exp\left(z_i^\top z_{p^*} / \tau\right)}{\exp\left(z_i^\top z_{p^*} / \tau\right) + \sum_{n \in \mathcal{N}^*(i)} \exp\left(z_i^\top z_n / \tau\right)}, \quad (3)$$

where $\mathcal{P}(i) = \{p, S(i)\}$ includes the original positive pair $p$ and the samples most similar in terms of domain features, whereas $\mathcal{N}_i^* = \mathcal{N}(i) \backslash S(i)$ denotes the negative sets excluding the updated positive samples. By drawing the representations of $i$ and $\mathcal{P}(i)$ closer together, we aim to guide the model to better align samples with meaningful semantic relationships.

**Domain knowledge guided prototype-level contrast**. Instance-level contrastive learning, particularly within each mini-batch, often fails to capture global semantic relationships adequately[40,51]. Several works[52,53] have incorporated group-level contrast beyond individual separation by unsupervised clustering of the embedding space. By ensuring that semantically similar samples remain clustered, such methods preserve both local and global semantic structures.

To extend this idea to healthcare applications, where domain knowledge plays a crucial role, we introduce domain knowledge-guided prototype contrastive learning. This method better preserves the local and global semantic relationships by leveraging domain-specific features during the contrastive learning process.

Offline prototype assignment. In the offline phase, we apply unsupervised k-means clustering on the domain features $\{d_i\}$ to derive $K$ clusters based

on their similarity. Each sample $x_i$ is assigned to the nearest cluster centroid $k_i$, reflecting similar domain feature proximity. This ensures that samples in the same cluster share domain-informed semantics, and vice versa. Subsequently, for each cluster, a prototype $\{c_k\}$ is derived and updated during training. We empirically set $K$ as 128. Further analysis of the number of prototypes is available in the Supplementary Material Section S4.3.

Online prototype update. The prototypes are updated online through an EMA of the cluster's mean representation within each mini-batch:

$$c_i' = \alpha c_i + (1 - \alpha)\frac{1}{n_i}\sum_j z_j \cdot \mathbb{1}\{k_j = i\}, \qquad (4)$$

where $\mathbb{1}$ is an indicator function that returns 1 when $k_j = i$, and 0 otherwise; $n_i$ denotes the number of samples that belong to cluster $i$ in each mini-batch; $\alpha$ is the momentum parameter (set as 0.5). This process ensures prototypes evolve as new representations are generated, while maintaining consistency with past observations[54,55].

Prototype-level contrast. Building on the online-updated prototypes, we apply prototype-level contrastive learning for each individual sample, ensuring that their representations align closely with their assigned prototypes while maintaining appropriate distances from other clusters. To optimize the learning process, we adjust temperatures dynamically based on the number of samples within each cluster, following previous work on prototype-based contrastive learning[52]. It introduced a dynamic temperature strategy, based on the observation that smaller temperatures would result in a larger concentration of the corresponding prototype. This dynamic temperature strategy ensures that the distribution of latent features around each prototype has similar concentration.

In particular, the temperature is defined as:

$$\tau_k = \frac{\sum_{i=1}^{n_k} ||z_i - c_k||_2}{n_k \log(n_k + m)}, \qquad (5)$$

where $n_k$ is the number of samples in cluster $k$, and $m$ is the smoothing parameter to avoid too large $\tau$ (empirically set as 10). They were initialized to the same value as $\tau$. Subsequently, they were updated after each epoch and normalized to have a mean of $\tau$. Further ablation analysis of this dynamic strategy is available in the Supplementary Material Section S4.3.

Finally, our domain knowledge-guided prototype-level contrastive learning is calculated as below,

$$\mathcal{L}_{con}^{proto}(x_i) = -\log\frac{\exp(z_i^\top c_{k_i}/\tau_i)}{\sum_{j=1}^K \exp(z_i^\top c_j/\tau_j)}. \qquad (6)$$

## Overall training
With the integration of both domain knowledge-guided instance and prototype-level learning, the whole pipeline of our framework is illustrated in Algorithm 1. Given that representations of prototypes are generally of lower quality in the initial training phases, we progressively applied prototypical contrastive learning. The final loss is formulated as

$$\mathcal{L} = \mathcal{L}_{con}^{ins} + \min\left(\max\left(\frac{T - T_0}{T_{max}}, 0\right), 1\right)\mathcal{L}_{con}^{proto}, \qquad (7)$$

where $T$, $T_0$, $T_{max}$ are the current epoch, the starting epoch for applying $\mathcal{L}_{con}^{proto}$, total epoch number, respectively.

## Experiment settings
The experiments were conducted using PyTorch on NVIDIA V100 GPUs. For the self-supervised training experiments, the models were trained for 1000 epochs with a batch size of 1024, using the Adam optimizer with an initial learning rate of 0.001 and a warm-up strategy for the first 5 epochs. Logistic regression was applied every 10 epochs on the validation set to monitor model selection. For transfer learning and semi-supervised learning, we reduced the learning rate to 0.0005 and limited the training to 40 epochs.

---

**Algorithm 1**. Domain Knowledge Guided Self-Supervised Contrastive Learning.
**Input:** Training Dataset $D$;
**Hyperparameters:** Epoch parameters $T_{max}$, $T_0$, cluster number K, temperature $\tau$;
**Initialized Model Parameters:** Feature Extractor (with projector) $f$;
**Output:** Optimized $f$.

```
1: Extract domain features {dᵢ}.
2:                                                    ▷ Offline Prototype Assignment.
3: Apply K-means on {dᵢ} to derive K clusters.
4:
5: for Epoch < Max Epochs do
6:     for t = 1 to Max Iterations do
7:         Sample {xᵢ, dᵢ}ᵢ₌₁ᴮ from D.
8:         Calculate zᵢ based on f(xᵢ).              ▷ Instance-Level Contrast.
9:         Calculate {𝒫(i)}.
10:        Calculate ℒᶜᵒⁿⁱⁿˢ.
11:
12:        Update prototype cₖ using EMA of zᵢ.       ▷ Online Prototype Update.
13:        Calculate inter-cluster distances.         ▷ Prototype-Level Contrast.
14:        Calculate {τₖ}.
15:        Calculate ℒᶜᵒⁿᵖʳᵒᵗᵒ.
16:        Update f based on ℒᶜᵒⁿⁱⁿˢ, ℒᶜᵒⁿᵖʳᵒᵗᵒ.
17:    end for
18:    Update {τₖ}.
19: end for
```

Given the imbalanced nature of most datasets, we employed Balanced Softmax Cross Entropy[56] to handle class imbalance. Without loss of generality, this was utilized for the training of all downstream experiments. Performance was assessed using the class-average F1 score. Data augmentation techniques, including temporal jittering, scaling, magnitude warping, Gaussian noise, and cutout, were applied to generate diverse views of the data following previous works[10,15]. Although some recent work has explored domain-specific augmentations for certain modalities (e.g., spatial transformations of ECG using vectorcardiography[17]), designing such strategies for diverse wearable signals is challenging and often requires modality-specific tailoring. In our study, we adopted a consistent, modality-agnostic augmentation pipeline to maintain simplicity and better highlight the effect of our domain-knowledge-guided contrastive learning framework. Further analysis is available in (Supplementary Section S3.1). Further implementation details are available in the Supplementary Material (Supplementary Sections S3.2 and S3.3).

## Data availability

All the datasets used in this work are from public resources, including MIMIC-III WDB, CinC17 (https://physionet.org/content/challenge-2017/1.0.0/), CPSC (http://2018.icbeb.org/Challenge.html), SleepEDF (https://www.physionet.org/content/sleep-edfx/1.0.0/), and Capture24 (https://github.com/OxWearables/capture24), subject to a satisfactory data usage agreement.

## Code availability

The preprocessing script and methodology development code for this study are available on https://github.com/guxiao0822/domainssl. Meanwhile, please refer to Supplementary Material Section S3 for more detailed guidance on implementations of the compared methods.

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

## Acknowledgements

D.A.C. was supported by the Pandemic Sciences Institute at the University of Oxford; the National Institute for Health Research (NIHR) Oxford Biomedical Research Centre (BRC); an NIHR Research Professorship; a Royal Academy of Engineering Research Chair; the Wellcome Trust-funded VITAL project (grant 204904/Z/16/Z); the EPSRC (grant EP/W031744/1); and the InnoHK Hong Kong Centre for Cerebro-cardiovascular Engineering (COCHE).

## Author contributions

X.G. conceived and designed the study, performed the data analysis, and drafted the manuscript. Z.L. conceived and designed the study and revised the manuscript. J.H. performed the data analysis and revised the manuscript. J.Q, W.F., L.L., L.C., Y.T.Z. contributed to the methodology design, result interpretation, and manuscript revision. D.A.C. conceived and supervised the project and revised the manuscript. All the authors contributed to the manuscript's finalization.

## Competing interests

The authors declare no competing interests.
