## [Transparent Peer Review file · Communications Engineering]

S1 Dataset Details

S1.1 CinC17

CinC17 is a dataset consisting of short ECG recordings collected from portable chest patches [1]. Each recording was annotated as Normal sinus rhythm, Atrial fibrillation (AF), Other abnormal rhythms, or Noisy recordings, by an outsourced annotation service.

All signals were sampled at 300 Hz. To standardize input length, we truncated or zero-padded each sequence to a fixed 10-second window. Bandpass filtering (0.5–40Hz) and z-normalization were applied to each segment.

S1.2 CPSC

This dataset [2] is comprised of standard 12-lead ECG recordings with cardiac abnormalities annotated by cardiologists. Following previous scenarios, we extracted lead II from the standard 12-lead ECGs to mimic wearable channels [3]. There are a total of nine different types of rhythm/morphology abnormalities provided in this dataset, including Normal sinus rhythm, Atrial fibrillation (AF), First-degree atrioventricular block (I-AVB), Left bundle branch block (LBBB), Right bundle branch block (RBBB), Premature atrial contraction (PAC), Premature ventricular contraction (PVC), ST-segment depression (STD), ST-segment elevation (STE). We extracted the primary annotation for each recording as the label, formulating as a multi-class classification problem.

ECG signals were sampled at 500 Hz. We truncated or zero-padded each signal to a fixed 10-second window and applied bandpass filtering (0.5–40 Hz) and z-normalization to standardize preprocessing across datasets.

S1.3 Simband

The UMass Simband dataset [4, 5] is a publicly available collection of wrist-worn photoplethysmography (PPG) signals acquired using the Samsung Simband 2.0 smartwatch. The dataset was designed to support research in wearable-based cardiac arrhythmia detection.

It comprises long-term PPG recordings from multiple participants, which we segmented into 10-second windows. Each segment was labeled as Normal sinus rhythm (NSR), Atrial fibrillation (AF), Premature atrial contraction (PAC)/Premature ventricular contraction (PVC), or Noise. Annotations were derived from synchronized 7-lead Holter ECG recordings, which served as the gold-standard reference, validated by a board-certified cardiologist.

Based on the official preprocessing implementations¹, they were segmented into 7590 10-second windows. Following this, bandpass filtering (0.4–30Hz) and z-normalization were subsequently applied.

S1.4 SleepEDF

The Sleep-EDF dataset [6] consists of overnight polysomnographic EEG recordings collected from healthy subjects and individuals with mild sleep difficulties. We selected two EEG channels (Fpz-Cz and Pz-Oz), which are commonly used in sleep staging research. Sleep stages were annotated for each 30-second window based on the standard scoring system by trained sleep technicians.

Each recording is segmented into 30-second non-overlapping epochs, resulting in 3,000 time steps per segment at a sampling rate of 100 Hz. We applied bandpass filtering (0.4–30 Hz) and z-normalization to each segment. The classification labels include five consolidated sleep stages: Wake (W), Rapid Eye Movement (REM), and non-REM stages N1, N2, and N3 (where N3 includes both Stage 3 and Stage 4).

S1.5 Capture24

The Capture24 dataset [7] is a free-living human activity recognition dataset collected in Oxfordshire, UK. The accelerometer data was sampled at 100 Hz with a dynamic range of $\pm 8g$. To obtain ground-truth activity labels, participants also wore a wearable camera that automatically captured images during waking hours. Activity labels were assigned based on the time-stamped wearable camera images and sleep diaries. Annotators were trained and quality-checked, achieving a high Cohen’s kappa score on a gold-standard dataset.

¹https://github.com/Cassey2016/UMass_Simband_Dataset

We picked up the first 20 subjects, with their sequence segmented into 10-second non-overlapping windows for analysis. We followed Willetts *et al.* [8], linking each with six high-level activity categories: bicycling, sit/stand, walking, vehicle, mixed activity, and sleep. We did not apply z-normalization to the original instances, as the absolute value range was meaningful and normalization would diminish its original magnitude characteristics.

S1.6 MIMIC-III-WDB

This dataset consists of ECG waveforms acquired from ICU bedside monitors, we used the matched database version. No segment-level cardiovascular disease-related annotations were provided within the original dataset, and it was mainly used for self-supervised training.

To curate training signals, we parsed all header files and selected those that contained both Lead II ECG signals. We excluded short or incomplete segments and retained only those with signal durations longer than 20 seconds. From each selected recording, we extracted the second 10-second segment and applied a series of signal quality checks. We used NeuroKit² to clean the signal and applied its ECG signal quality index for assessment. Signals labeled as "Unacceptable" were discarded. Only segments passing all quality checks were retained. All selected segments were subsequently bandpass (0.5-40Hz) filtered and z-normalized.

S2 Domain Expertise Informed Features

We followed previously established work to preprocess the data and extract domain-knowledge-informed features, with the corresponding feature extraction scripts provided with the corresponding work. The details are provided in Tables S1, S2, S3 and S4, respectively.

Table S1. Domain expertise informed features of ECG [9].

AF-Related Feature	AF-Evidence [10], Shannon entropy [11], K-S test [12], Radius [9]
Morphological Features	QRS duration, PR interval, QT interval, QS interval, ST amplitude, P amplitude, Q amplitude, R amplitude, S amplitude, T amplitude.
RR Interval Related Feature	Median RR interval, Arrhythmia index [13]
Beat Similarity Related Feature	QRS similarity index, ratio of high similarity beats, R amplitude similarity index, ratio of noisy beats.

*We applied two different toolboxes to perform characteristic waveform PQRST detection, thus the number of features is doubled in total.

Table S2. Domain expertise informed features of EEG* [14].

Time Statistical Feature	Interquartile Range (IQR), Kurtosis, Skewness, Standard Deviation (Std), Zero-crossing
Time Entropy	Binned Entropy (5, 10, 30, 60 bins), Permutation Entropy
Time Hjorth Parameter	Complexity, Mobility
Time Fractal Dimension Feature	Petrosian, Higuchi
Frequency Statistical Feature	Spectral Centroid, Kurtosis, Skew, Std
Frequency Entropy	Fourier Binned Entropy (2, 3, 5, 10, 30, 60, 100 bins)
Frequency Band Power Feature	Total Power, Band Power (alpha, beta, fast delta, slow delta, sigma, theta), Band Power Ratio (fast delta+theta, alpha/theta, delta/beta, delta/sigma, delta/theta)

*We calculated the features for each channel, so the total number of features is doubled.

²<https://github.com/neuropsychology/NeuroKit>

Table S3. Domain expertise informed features of IMU [7].

Time Feature	Minimum, Maximum, Median, 25th and 75th percentiles of each axis and the integrated magnitude; correlations between each axis and 1-second lag autocorrelation of the magnitude stream
Angular Feature	Dynamic Roll, Pitch, Yaw (Mean and Std); Gravity Roll, Pitch, Yaw (Mean)
Spectral Feature	Spectral Power; first and second dominant frequencies and corresponding band powers
Peak Feature	Number of Peaks, Median Prominence of Peaks.

Table S4. Domain expertise informed features of PPG [15].

Interval Features	Pulse interval (time between pulse onset and offset), Peak-to-peak interval (time between two systolic peaks), Systolic time (from pulse onset to dicrotic notch), Diastolic time (from dicrotic notch to pulse offset), Systolic peak time (from pulse onset to systolic peak), Diastolic peak time (from pulse onset to diastolic peak), Time delay between systolic and diastolic peaks, Systolic width at 10/25/33/50/66/75/90% amplitude between pulse onset and systolic peak, Diastolic width at 10/25/33/50/66/75/90% amplitude between systolic peak and pulse offset, Pulse width (sum of systolic and diastolic widths at 10/25/33/50/66/75/90%), Instantaneous pulse rate, u/v/w/a/b/c/d/e/f/p1/p2 to onset interval, b to c/d interval, p1/p2 to diastolic peak interval.
Amplitude Feature	Systolic peak amplitude (from pulse onset to systolic peak), Dicrotic notch amplitude (from pulse onset to dicrotic notch), Diastolic peak amplitude (from pulse onset to diastolic peak), Pulse onset-to-offset amplitude, Amplitudes at u, v, w, a, b, c, d, e, f, p1, p2 points, Amplitude ratios (e.g. , u to systolic peak, v to u, etc.), Reflection index (diastolic to systolic peak amplitude), Aging indices (including original, modified AGI, and informal AGI), Augmentation index, Reflection indices for p1 and p2
Curve Area Feature	Area under the full pulse wave (onset to offset), Area under the systolic curve (onset to dicrotic notch), Area under the diastolic curve (dicrotic notch to offset), Inflection point area, Area-based ratios (e.g. , diastolic vs. systolic).
Combined Feature	Ratios between systolic and diastolic times, Ratios of pulse width to pulse interval, Ratios of pulse width to systolic peak time, Ratios of systolic peak time to pulse interval, Amplitude-to-time ratios (e.g. , systolic amplitude to time delay), Stiffness index, Ratio of systolic amplitude to difference between pulse interval and systolic peak time, Spring constant, Inflection point area plus normalized d-point amplitude.

S2.1 Open-Source Toolboxes

Furthermore, we also provide a curated list of widely-used, community-supported signal processing toolkits for different wearable sensing modalities, which may serve as a useful reference for future work in this area. The full list is provided in Table S5.

Table S5. Representative open-source toolkits for wearable signal preprocessing and feature extraction, along with supported wearable sensing modalities. They include ECG, PPG, EEG, IMU, RSP (Respiratory), ABP (Arterial Blood Pressure), EDA (Electrodermal Activity), MEG (Magnetoencephalography), fNIRS (functional Near-Infrared Spectroscopy).

Toolbox / Package	ECG	PPG	EEG	IMU	RSP	Others
WFDB Toolbox ¹	✓	✓	✓		✓	ABP, EMG
NeuroKit2 ²	✓	✓			✓	EDA , EMG
HeartPy ³	✓	✓				
BioSPPy ⁴	✓	✓	✓		✓	EDA, EMG, PCG
pyHRV ⁵	✓	✓				
PyPhysio ⁶	✓	✓		✓		EDA, fNIRS
BIOBSS ⁷	✓	✓		✓		EDA
MNE-Python ⁸			✓			MEG , ECoG, fNIRS
EEGLAB ⁹			✓			MEG, ECoG
PhysioKit ¹⁰		✓			✓	EDA
Vital_Sqi ¹¹	✓	✓				
PPGFeat ¹²		✓				

¹ <https://physionet.org/content/wfdb-matlab/0.10.0/>

² <https://github.com/neuropsychology/NeuroKit>

³ <https://python-heart-rate-analysis-toolkit.readthedocs.io/en/latest/>

⁴ <https://github.com/scientisst/BioSPPy>

⁵ <https://pyhrv.readthedocs.io/en/latest/>

⁶ <https://gitlab.com/a.bizzego/pyphysio>

⁷ <https://github.com/obss/BIOBSS>

⁸ <https://mne.tools/stable/>

⁹ <https://sccn.ucsd.edu/eeglab/>

¹⁰ <https://github.com/PhysiologicAIIab/PhysioKit>

¹¹ https://github.com/meta00/vital_sqi

¹² <https://github.com/saadur/PPGFeat>

S3 Experimental Details

S3.1 Augmentations

We implemented a series of five time-domain augmentations. They were applied to raw signals successively during training. These augmentations are commonly adopted in physiological signal processing and are tailored to simulate realistic variations in wearable time series data:

S3.1.1 Types

Temporal Jittering. Shifts the entire time series forward or backward by a random number of time steps, with a maximum shift of ± 5 time steps.

Temporal Scaling. Stretches or compresses the time axis of the signal by a random factor $[0.8, 1.2]$ within a defined range. The signal was subsequently resampled to its original length using cubic interpolation.

Magnitude Warping. A global scaling factor was randomly sampled from the range $[0.8, 1.2]$ and applied uniformly across all time points.

Additive Gaussian Noise. Zero-mean Gaussian noise with standard deviation $\sigma = 0.2$ was added to each time point.

Time Series Cutout. A contiguous segment amounting to 25% of the total time steps was randomly masked (set to zero).

S3.1.2 Effects of Augmentations

It is noted that the augmentations applied in this study are generic time series augmentation strategies. For simplicity and without loss of generality, we apply the same set of augmentations across different sensing modalities to assess the generalizability of our framework. Recently, modality-specific augmentation strategies have been explored, such as domain-aware transformations for inertial measurement units (IMUs) [16], or spatial augmentations for EEG signals [17]. Despite the effectiveness demonstrated, such approaches are relatively recent and often require careful,

modality-specific design. In contrast, domain-specific manual feature engineering has undergone years of refinement and validation, supported by mature, well-documented toolboxes, open-source packages, and publicly available repositories. Our work leverages this foundation to inject domain expertise into the self-supervised learning process in a reliable and scalable manner. In this sense, the use of generic time series augmentations does not compromise our key contribution of integrating domain expertise into SSL.

To assess the impact of individual augmentations on representation quality, we conducted ablation experiments using both supervised learning and self-supervised contrastive learning (SimCLR). Each augmentation was applied individually and in combination, and we measured downstream performance to evaluate their contribution to robustness against signal distortions. The results are presented in Table S6. As demonstrated, these augmentations improve representation robustness, as evidenced by performance gains on downstream tasks.

Table S6. Performance of individual and combined augmentations in supervised and self-supervised learning (SimCLR) on CPSC (ECG) dataset. The performance is reported by class-average F1.

Jittering	Scaling	Warping	Noise	Cutout	Supervised	Self-Supervised
✓					0.561	0.488
	✓				0.558	0.483
		✓			0.557	0.479
			✓		0.553	0.481
				✓	0.556	0.486
✓	✓				0.574	0.497
✓	✓	✓			0.581	0.508
✓	✓	✓	✓		0.588	0.523
✓	✓	✓	✓	✓	0.592	0.532

S3.2 Model architectures

Backbone Architecture. We implemented the basic feature extraction network using PyTorch, with one-dimensional convolutional backbones. Without specific mention, we applied ResNet1D-18 as the feature extractor. Specifically, we also applied other ResNet1D variants (ResNet1D-34, -50) and a multi-scale dense neural network (MSDNN) as the feature extractors, to test the model-agnostic nature of our method.

Projection Head. For contrastive learning, a two-layer projection head was added on top of the backbone: a linear layer mapping to a 128-dimensional space, followed by a ReLU activation, and another linear layer (128 to 128).

Downstream Classifier. A separate linear classification head was appended to the frozen or fine-tuned backbone for downstream tasks. It consists of a single linear layer projecting from the backbone’s output feature size to the number of classes.

S3.3 Experimental settings

Pretraining. All models were trained using the Adam optimizer with a learning rate of 0.001 and a batch size of 1024. The temperature parameter for the contrastive loss was set as $\tau = 0.07$, following standard SimCLR [18]. Models were trained for 1000 epochs with early stopping based on downstream validation performance. Logistic regression was applied every 10 epochs on the validation set to monitor model selection. The starting point for applying $\mathcal{L}_{con}^{proto}$ was empirically set as 40. All experiments were conducted using five random seeds (1,30,42,123,1234), with average performance reported.

Downstream. For transfer learning and semi-supervised learning, another randomly initialized classifier was added on top of the backbone. We reduced the learning rate to 0.0005 and limited the training to 40 epochs. For linear probing, similarly a randomly initialized classifier was applied with the learning rate as 0.001 and training epochs as 40.

S3.4 Implementation of other compared methods

For the pretraining of all other compared methods, the details are described as below. To be specific, the backbone and projector are set the same as ours for fair comparison.

SimCLR [18]: A baseline contrastive learning framework using instance discrimination and a temperature-scaled InfoNCE loss, with the same temperature settings as ours. All the following implementations used in this study were mostly modified and extended based on this baseline framework.

BYOL [19]: A bootstrap-based framework without negative pairs. We followed the same hyperparameter settings as in the official implementation³.

MoCo [20]: Momentum contrast learning with a dynamic memory bank. The backbone and projector is the same as SimCLR. We followed the same hyperparameter settings as in the official implementation⁴.

SwAV [21]: A prototype-based method with online clustering strategy. We followed the settings in the official implementation⁵.

NNCLR [22]: Uses nearest-neighbor matching to select positive pairs. We adopted the same matching strategy as in the official implementation⁶ as our version.

TS [23] A temperature schedule strategy proposed to tackle imbalanced data, especially long-tailed, during self-supervised learning. We followed the official implementation⁷.

AMCL [24] An adaptive multi-head contrastive learning framework. It modified the constant-temperature, single-head approach and to adaptive temperature, multi-head approaches. We followed the official implementation⁸.

CLOCS [25]: Contrastive learning on cardiac signals with cross-view pretext tasks. Following the official implementation⁹, we built on top of SimCLR, by performing contrastive learning across time and patients.

TFC [26]: A temporal and frequency-domain contrastive framework. We followed the official implementation¹⁰.

SoftIns [27]: A soft contrastive learning framework that introduces both instance-wise and time-wise soft contrastive objectives to enhance representation learning. Due to architectural limitations of our ResNet1D-18 backbone, which is not well-suited for fine-grained time-step-level supervision, we implemented only the instance-level soft contrastive learning component. It should be noted standalone instance-level soft contrastive also demonstrates performance gain as reported in [27]. Our implementation follows the official repository¹¹.

RNC [28]: The original Rank-and-Contrastive (RNC) loss was proposed for supervised contrastive learning in regression tasks. It ranks feature distances according to the distance between continuous target values and encourages the learned representations to reflect this order. To enable a comprehensive comparison in terms of leveraging domain features for self-supervised learning, we adapted official RNC implementation¹² by ranking the pairwise distances between domain-specific feature vectors instead of regression targets. This allows the RNC framework to operate without labels.

MOMENT [20]: MOMENT is a general purpose time series foundation model, with both its official implementation and pretrained model publicly available¹³. We followed the official guidelines and used “MOMENT-1-large” version to extract representations.

Chronos [29]: Chronos is another general purpose time series foundation model, with open-source code and pretrained weights accessible online¹⁴. We employed the “t5-base” version for representation extraction. Notably, since Chronos supports only single-channel time series, we applied it separately to each channel and then averaged the resulting representations across channels to obtain the final feature vector.

S4 Other Results

S4.1 Comparison with Wearable-Modality Specific Foundation Models

We additionally include results comparing our method against modality-specific pretrained models designed for waveforms [30, 31]. While such models are often tuned for particular sensor types or clinical tasks and thus may not

³<https://github.com/facebookresearch/moco>

⁴<https://github.com/facebookresearch/moco>

⁵<https://github.com/facebookresearch/swav>

⁶<https://github.com/MalteEbner/NNCLR>

⁷https://github.com/Annusha/temperature_schedules

⁸<https://github.com/LeiWangR/cl>

⁹<https://github.com/danikiyasseh/CLOCS>

¹⁰<https://github.com/mims-harvard/TFC-pretraining>

¹¹<https://github.com/seunghan96/softclt>

¹²<https://github.com/kaiwenzha/Rank-N-Contrast>

¹³<https://github.com/moment-timeseries-foundation-model/moment>

¹⁴<https://github.com/amazon-science/chronos-forecasting>

generalize across modalities, we nevertheless evaluated them to gain additional insight..

In detail, we compared with two wearable-modality-specific foundation models (ECGFM [30] for 12-Lead Diagnostic ECG and PaPaGei [31] for PPG). The results, measured in terms of class-average F1 score, are summarized below, with our model pretrained using self-supervised learning.

Table S7. Comparison with wearable-modality-specific foundation models. We evaluate our self-supervised pretrained model against ECGFM [30] and PaPaGei [31], two foundation models tailored for 12-lead ECG and PPG signals, respectively. Performance is reported as class-average F1 score using both KNN and linear probing.

Method	KNN			Linear Probing		
	ECG (CPSC)	ECG (CinC17)	PPG (SimBand)	ECG (CPSC)	ECG (CinC17)	PPG (SimBand)
PaPaGei[31]	-	-	0.335	-	-	0.417
ECGFM[30]	0.511	0.565	-	0.557	0.582	-
Ours	0.509	0.571	0.420	0.574	0.540	0.499

For ECGFM, we used the publicly available pretrained model¹⁵ and adapted our data by duplicating the single-channel ECG signals to 12 channels to match the model’s input requirements. For PaPaGei, we followed the official implementation pipeline¹⁶ for PPG representation extraction and evaluation.

As shown in Table S7, ECGFM performs competitively on ECG datasets and occasionally surpasses our method, particularly on CinC17. On the other hand, PaPaGei also shows reasonable performance on PPG signals.

Please note that ECGFM was pretrained on the CPSC dataset in a self-supervised manner, and since the exact pretraining setup is not fully disclosed, it is possible that test data may have been included during pretraining.

It is also important to note that direct comparisons with these large foundation models are not entirely equitable, due to differences in model size, training scale, and modality-specific design. Foundation models are typically advantageous due to their zero-shot generalization capability. Nevertheless, incorporating our domain-guided strategies into future foundation model development may enhance both their interpretability and clinical applicability.

S4.2 Performance of Minority Categories

As shown in Figure S1, we plot the per-class accuracy improvement of our method compared to SimCLR, against class size (sorted from most to least frequent class). The black line indicates the number of training samples per class, while the green bars represent the change in classification accuracy achieved by our approach.

This analysis reveals that our method offers more stable and often larger improvements for classes with fewer training samples, whereas performance gains for more common classes are more variable. This result confirms that domain knowledge guided contrast can be especially beneficial for underrepresented classes, often the ones associated with rare but critical clinical conditions.

S4.3 Other Ablation Studies

We also conducted several ablation studies on implementation choices that, while not central to our key contributions, provide further insights into design decisions.

Temperature strategy. Our prototype-based contrastive learning approach draws inspiration from prior work [32]. Especially, there are two implementation details: (1) assigning a cluster-specific temperature to each prototype, and (2) progressively introducing the prototype-level contrastive loss during training. As shown in Table S8, using a cluster-specific temperature yields better performance, likely due to the improved adaptability across clusters with varying intra-group variance (comparing **a** with **e**). On the other hand, applying prototype-level contrastive learning together with instance-level contrastive learning from the start leads to suboptimal results. This might be because the prototype representations at the very initial point may not be good enough (only updated with a small number of samples). Forcing alignment of each instance to their positive pair and the corresponding prototype too early could destabilize training and hinder convergence (comparing **b** with **e**).

¹⁵<https://github.com/bowang-lab/ECG-FM>

¹⁶<https://github.com/bowang-lab/ECG-FM>

Figure S1. Per-class accuracy improvement compared to SimCLR, plotted against class size (sorted from largest to smallest). The line represents the number of training samples per class, while bars indicate the change in accuracy achieved by our method over SimCLR. Classes with smaller sample sizes generally exhibit more stable improvements, while gains in larger classes are more variable.

Number of clusters. Additionally, we explored the impact of varying the number of clusters k during training within our whole framework. As shown in Table S8 (comparing **d,e,f**), our method remains robust to changes in k , exhibits consistent performance across a range of cluster sizes. Based on empirical observations, we selected $k = 128$ to achieve a relatively fine-grained clustering and model efficiency.

Table S8. Additional experiments on CPSC (ECG) dataset. This includes ablation studies of different framework components and sensitivity studies of the clustering number.

Version	F1
a w/o temperature-wise schedule	0.563
b w/o progressive	0.522
d # Proto $k=32$	0.569
e # Proto $k=128$ (Ours)	0.574
f # Proto $k=256$	0.572

S5 Extended Related Work

S5.1 Curation of Healthcare Wearable Datasets

Annotating the complex waveforms from healthcare wearables requires prohibitively high time, cost, and expertise. Annotation schemes for wearable data generally fall into two main categories:

Direct Manual Annotation from Raw Data. Expert clinicians manually inspect and annotate waveforms, which is labor-intensive, time-consuming, and requires specialized medical knowledge. For instance, annotating ECG or EEG recordings typically involves cardiologists or neurologists, respectively, carefully labeling abnormalities or distinct waveform patterns. Achieving consistent and accurate annotations generally necessitates multiple senior clinicians reviewing each recording, and to reach consensus, further increasing resource demands [33].

Annotation Using Additional “Gold-Standard” Devices. Although raw waveforms from wearables are rich in information, obtaining explicit annotations often requires external reference equipment to establish reliable ground-truth labels. For instance, accurately annotating activity labels typically requires using external cameras as a reference, mostly coupled with crowdsourcing or expert labeling, to generate dependable ground-truth annotations. Notable

examples include datasets like Capture24 [7], where wearable signals are aligned and annotated using synchronized video recordings, to obtain reliable ground-truth data. Annotators typically require systematic training, and achieving consistent, reliable consensus among multiple annotators remains essential [7].

S5.2 Learning with Healthcare Wearables with Minimal Supervision

Faced with the challenges of annotating the waveforms streamed from wearables, emerging direction in healthcare wearables is to perform label-efficient decoding of healthcare wearables, minimizing the need for labels whilst learning good representations from these high-dimensional waveforms [34].

Self-Supervised Learning. Self-supervised learning (SSL) has become a dominant strategy for label-efficient learning in healthcare wearables. Among SSL methods, contrastive learning has been particularly popular [18]. It trains models to pull together representations of positive pairs (typically different augmented views of the same instance) while pushing apart negative pairs (different instances). This discriminative pretraining objective has been shown to encourage the model to learn meaningful and generalizable feature representations without manual supervision. However, as discussed in the main paper, these conventional instance-level contrast may not learn semantically meaningful representations.

In healthcare wearables, (or more generally, the time series domain), several advances have been made in self-supervised contrastive learning, identified across segment-, temporal-, individual-, and medical concept-level strategies. At the segment level, much of the focus has been on curating domain-specific/generic augmentations for waveform signals. Representative examples include optimizing the temporal-frequency representation consistency in [26], and some domain-specific augmentations [17]. Given the inherently sequential nature of biosignals, several methods further exploit temporal relationships [35, 27] to define contrastive views. For instance, Lan *et al.* [35] introduced a stationarity-aware contrastive learning strategy that detects and preserves abrupt changes across neighboring frames. However, these approaches typically require access to long continuous recordings, which are not always available in wearable datasets collected under specific paradigms. Moreover, within-subject sequences can span varying physiological or pathological states, complicating identity-based consistency.

Beyond segments and timestamps, individual-level contrastive learning aims to preserve subject-specific characteristics across recordings. Sangha *et al.* [36] and Abbaspourazad *et al.* [37] showed that identity-aware contrastive learning improves representation quality and outperforms naive segment-level strategies. Still, this line of work assumes that multiple recordings are available per subject, which is often not the case in wearables. Lastly, some recent efforts have incorporated external modalities, such as paired clinical notes, to guide contrastive representation learning. Representative examples include recent ECG-Text contrastive learning framework proposed by Yu *et al.* [38]. Such approaches, while promising, rely on the availability of well-aligned auxiliary data, which many wearable modalities may lack, limiting their general applicability.

Time Series Foundation Models. Recently, self-supervised learning strategies have been extended to the training of large-scale models on diverse and heterogeneous time series datasets, giving rise to time series foundation models [39]. These models aim to learn general-purpose representations that can be transferred across a wide range of downstream tasks and domains. In the broader time series domain, several foundation models have been proposed [40, 41, 29], trained on large-scale corpora spanning domains such as finance, weather, traffic, etc. However, most of these pretraining datasets cover little to no medical (or even healthcare wearable) data. Given the fundamental differences in dynamics, noise characteristics, and clinical semantics between healthcare waveforms and non-medical time series, the applicability and effectiveness of these general-purpose foundation models in healthcare wearables remains largely unexplored and unvalidated.

On the other hand, several wearable-modality-specific foundation models have recently been proposed for wearable signals, such as PaPaGei [31] and Accelerometer Foundation Model [16]. These models have demonstrated strong representation learning capabilities on unseen data within the same modality. However, their pretraining paradigms often rely on modality-specific designs [16, 31] as the key contributions, limiting their applicability across diverse wearable sensing modalities.

That said, the development of full-scale foundation models is beyond the scope of this work. Instead, our focus is on advancing self-supervised contrastive learning strategies that are inherently generalizable across different wearable modalities. Importantly, we aim to provide practical guidance on how “readily available” medical knowledge can be incorporated into the self-supervised pretraining process, for training wearable foundation models, which is one of our future directions.

Semi-Supervised Learning. Compared to self-supervised learning, semi-supervised learning offers another promising strategy for label-efficient decoding of healthcare wearables by leveraging a small amount of labeled data alongside a

large pool of unlabeled signals. A straightforward and simplistic approach is to integrate supervised training on the labelled subset, coupled with self-supervised learning over the whole set. Compared to self-supervised learning, more advanced strategies incorporate some surrogate objectives, such as pseudo-labeling or consistency regularization, to allow the model to benefit from both labeled guidance and unlabelled training signals.

These have been applied in wearable sensing modalities. Pseudo-labeling is a technique in which model predictions on unlabeled data are treated as temporary ground-truth labels for training. The key idea has been applied to semi-supervised learning of ECG [42] and EEG [43] signals. Another popular direction is consistency regularization, which assumes that model predictions should remain stable under input perturbations or augmentations. Among these, FixMatch [44], a widely used method, applies weak and strong augmentations to the same input and trains the model to produce consistent predictions. This has been applied and extended in wearable contexts like accelerometer based activity recognition [45].

References

- [1] Clifford GD, Liu C, Moody B, Li-wei HL, Silva I, Li Q, et al. AF classification from a short single lead ECG recording: The PhysioNet/computing in cardiology challenge 2017. In: 2017 Computing in Cardiology (CinC). IEEE; 2017. p. 1-4.
- [2] Liu F, Liu C, Zhao L, Zhang X, Wu X, Xu X, et al. An open access database for evaluating the algorithms of electrocardiogram rhythm and morphology abnormality detection. *Journal of Medical Imaging and Health Informatics*. 2018;8(7):1368-73.
- [3] Hannun AY, Rajpurkar P, Haghpanahi M, Tison GH, Bourn C, Turakhia MP, et al. Cardiologist-level arrhythmia detection and classification in ambulatory electrocardiograms using a deep neural network. *Nature medicine*. 2019;25(1):65-9.
- [4] Bashar SK, Han D, Hajeb-Mohammadalipour S, Ding E, Whitcomb C, McManus DD, et al. Atrial fibrillation detection from wrist photoplethysmography signals using smartwatches. *Scientific reports*. 2019;9(1):15054.
- [5] Han D, Bashar SK, Mohagheghian F, Ding E, Whitcomb C, McManus DD, et al. Premature atrial and ventricular contraction detection using photoplethysmographic data from a smartwatch. *Sensors*. 2020;20(19):5683.
- [6] Kemp B, Zwinderman AH, Tuk B, Kamphuisen HA, Obery JJ. Analysis of a sleep-dependent neuronal feedback loop: the slow-wave microcontinuity of the EEG. *IEEE Transactions on Biomedical Engineering*. 2000;47(9):1185-94.
- [7] Chan S, Yuan H, Tong C, Acquah A, Schonfeldt A, Gershuny J, et al.. CAPTURE-24: A large dataset of wrist-worn activity tracker data collected in the wild for human activity recognition; 2024.
- [8] Willetts M, Hollowell S, Aslett L, Holmes C, Doherty A. Statistical machine learning of sleep and physical activity phenotypes from sensor data in 96,220 UK Biobank participants. *Scientific reports*. 2018;8(1):7961.
- [9] Shao M, Bin G, Wu S, Bin G, Huang J, Zhou Z. Detection of atrial fibrillation from ECG recordings using decision tree ensemble with multi-level features. *Physiological measurement*. 2018;39(9):094008.
- [10] Sarkar S, Ritscher D, Mehra R. A detector for a chronic implantable atrial tachyarrhythmia monitor. *IEEE Transactions on Biomedical Engineering*. 2008;55(3):1219-24.
- [11] Dash S, Chon K, Lu S, Raeder E. Automatic real time detection of atrial fibrillation. *Annals of biomedical engineering*. 2009;37:1701-9.
- [12] Huang C, Ye S, Chen H, Li D, He F, Tu Y. A novel method for detection of the transition between atrial fibrillation and sinus rhythm. *IEEE Transactions on Biomedical Engineering*. 2010;58(4):1113-9.
- [13] Tsipouras MG, Fotiadis DI, Sideris D. An arrhythmia classification system based on the RR-interval signal. *Artificial intelligence in medicine*. 2005;33(3):237-50.

- [14] Van Der Donckt J, Van Der Donckt J, Deprost E, Vandenbussche N, Rademaker M, Vandewiele G, et al. Do not sleep on traditional machine learning: Simple and interpretable techniques are competitive to deep learning for sleep scoring. *Biomedical Signal Processing and Control*. 2023;81:104429.
- [15] Goda MÁ, Charlton PH, Behar JA. pyPPG: a Python toolbox for comprehensive photoplethysmography signal analysis. *Physiological Measurement*. 2024;45(4):045001.
- [16] Yuan H, Chan S, Creagh AP, Tong C, Acquah A, Clifton DA, et al. Self-supervised learning for human activity recognition using 700,000 person-days of wearable data. *NPJ digital medicine*. 2024;7(1):91.
- [17] Gopal B, Han R, Raghupathi G, Ng A, Tison G, Rajpurkar P. 3KG: Contrastive learning of 12-lead electrocardiograms using physiologically-inspired augmentations. In: *Machine learning for health*. PMLR; 2021. p. 156-67.
- [18] Chen T, Kornblith S, Norouzi M, Hinton GE. A Simple Framework for Contrastive Learning of Visual Representations. In: *Proceedings of the 37th International Conference on Machine Learning, ICML 2020, 13-18 July 2020, Virtual Event*. vol. 119 of *Proceedings of Machine Learning Research*. PMLR; 2020. p. 1597-607. Available from: <http://proceedings.mlr.press/v119/chen20j.html>.
- [19] Grill J, Strub F, Altché F, Tallec C, Richemond PH, Buchatskaya E, et al. Bootstrap Your Own Latent - A New Approach to Self-Supervised Learning. In: Larochelle H, Ranzato M, Hadsell R, Balcan M, Lin H, editors. *Advances in Neural Information Processing Systems 33: Annual Conference on Neural Information Processing Systems 2020, NeurIPS 2020, December 6-12, 2020, virtual*; 2020. Available from: <https://proceedings.neurips.cc/paper/2020/hash/f3ada80d5c4ee70142b17b8192b2958e-Abstract.html>.
- [20] He K, Fan H, Wu Y, Xie S, Girshick RB. Momentum Contrast for Unsupervised Visual Representation Learning. In: *2020 IEEE/CVF Conference on Computer Vision and Pattern Recognition, CVPR 2020, Seattle, WA, USA, June 13-19, 2020*. IEEE; 2020. p. 9726-35. Available from: <https://doi.org/10.1109/CVPR42600.2020.00975>.
- [21] Caron M, Misra I, Mairal J, Goyal P, Bojanowski P, Joulin A. Unsupervised Learning of Visual Features by Contrasting Cluster Assignments. In: Larochelle H, Ranzato M, Hadsell R, Balcan M, Lin H, editors. *Advances in Neural Information Processing Systems 33: Annual Conference on Neural Information Processing Systems 2020, NeurIPS 2020, December 6-12, 2020, virtual*; 2020. Available from: <https://proceedings.neurips.cc/paper/2020/hash/70feb62b69f16e0238f741fab228fec2-Abstract.html>.
- [22] Dwibedi D, Aytar Y, Tompson J, Sermanet P, Zisserman A. With a Little Help from My Friends: Nearest-Neighbor Contrastive Learning of Visual Representations. In: *2021 IEEE/CVF International Conference on Computer Vision, ICCV 2021, Montreal, QC, Canada, October 10-17, 2021*. IEEE; 2021. p. 9568-77. Available from: <https://doi.org/10.1109/ICCV48922.2021.00945>.
- [23] Kukleva A, Böhle M, Schiele B, Kuehne H, Rupperecht C. Temperature Schedules for self-supervised contrastive methods on long-tail data. In: *The Eleventh International Conference on Learning Representations, ICLR 2023, Kigali, Rwanda, May 1-5, 2023*. OpenReview.net; 2023. Available from: <https://openreview.net/pdf?id=ejHUr4nfHhD>.
- [24] Wang L, Koniusz P, Gedeon T, Zheng L. Adaptive multi-head contrastive learning. In: *European Conference on Computer Vision*. Springer; 2024. p. 404-21.
- [25] Kiyasseh D, Zhu T, Clifton DA. CLOCS: Contrastive Learning of Cardiac Signals Across Space, Time, and Patients. In: Meila M, Zhang T, editors. *Proceedings of the 38th International Conference on Machine Learning, ICML 2021, 18-24 July 2021, Virtual Event*. vol. 139 of *Proceedings of Machine Learning Research*. PMLR; 2021. p. 5606-15. Available from: <http://proceedings.mlr.press/v139/kiyasseh21a.html>.
- [26] Zhang X, Zhao Z, Tsiligkaridis T, Zitnik M. Self-Supervised Contrastive Pre-Training For Time Series via Time-Frequency Consistency. In: Koyejo S, Mohamed S, Agarwal A, Belgrave D, Cho K, Oh A, editors. *Advances in Neural Information Processing Systems 35: Annual Conference on Neural Information Processing Systems 2022, NeurIPS 2022, New Orleans, LA, USA, November 28 - December 9, 2022*; 2022. Available from: http://papers.nips.cc/paper_files/paper/2022/hash/194b8dac525581c346e30a2cebe9a369-Abstract-Conference.html.

- [27] Lee S, Park T, Lee K. Soft Contrastive Learning for Time Series. In: The Twelfth International Conference on Learning Representations; 2024. Available from: <https://openreview.net/forum?id=pAsQSW1DUf>.
- [28] Zha K, Cao P, Son J, Yang Y, Katabi D. Rank-N-Contrast: Learning Continuous Representations for Regression. In: Oh A, Naumann T, Globerson A, Saenko K, Hardt M, Levine S, editors. Advances in Neural Information Processing Systems 36: Annual Conference on Neural Information Processing Systems 2023, NeurIPS 2023, New Orleans, LA, USA, December 10 - 16, 2023; 2023. Available from: http://papers.nips.cc/paper_files/paper/2023/hash/39e9c5913c970e3e49c2df629daff636-Abstract-Conference.html.
- [29] Ansari AF, Stella L, Turkmen AC, Zhang X, Mercado P, Shen H, et al. Chronos: Learning the Language of Time Series. Transactions on Machine Learning Research. 2024. Expert Certification. Available from: <https://openreview.net/forum?id=gerNCVqqtR>.
- [30] McKen K, Oliva L, Masood S, Toma A, Rubin B, Wang B. Ecg-fm: An open electrocardiogram foundation model. arXiv preprint arXiv:240805178. 2024.
- [31] Pillai A, Spathis D, Kawsar F, Malekzadeh M. PaPaGei: Open Foundation Models for Optical Physiological Signals. In: The Thirteenth International Conference on Learning Representations; 2025. Available from: <https://openreview.net/forum?id=kYwTmlq6Vn>.
- [32] Li J, Zhou P, Xiong C, Hoi SCH. Prototypical Contrastive Learning of Unsupervised Representations. In: 9th International Conference on Learning Representations, ICLR 2021, Virtual Event, Austria, May 3-7, 2021. OpenReview.net; 2021. Available from: <https://openreview.net/forum?id=KmykpuSrjccq>.
- [33] Lai J, Tan H, Wang J, Ji L, Guo J, Han B, et al. Practical intelligent diagnostic algorithm for wearable 12-lead ECG via self-supervised learning on large-scale dataset. Nature Communications. 2023;14(1):3741.
- [34] Gu X, Deligianni F, Han J, Liu X, Chen W, Yang GZ, et al. Beyond supervised learning for pervasive healthcare. IEEE Reviews in Biomedical Engineering. 2023.
- [35] Lan X, Ng D, Hong S, Feng M. Intra-Inter Subject Self-Supervised Learning for Multivariate Cardiac Signals. In: Thirty-Sixth AAAI Conference on Artificial Intelligence, AAAI 2022, Thirty-Fourth Conference on Innovative Applications of Artificial Intelligence, IAAI 2022, The Twelfth Symposium on Educational Advances in Artificial Intelligence, EAAI 2022 Virtual Event, February 22 - March 1, 2022. AAAI Press; 2022. p. 4532-40. Available from: <https://ojs.aaai.org/index.php/AAAI/article/view/20376>.
- [36] Sangha V, Khunte A, Holste G, Mortazavi BJ, Wang Z, Oikonomou EK, et al. Biometric contrastive learning for data-efficient deep learning from electrocardiographic images. Journal of the American Medical Informatics Association. 2024;31(4):855-65.
- [37] Abbaspourazad S, Elachqar O, Miller A, Emrani S, Nallasamy U, Shapiro I. Large-scale Training of Foundation Models for Wearable Biosignals. In: The Twelfth International Conference on Learning Representations; 2024. Available from: <https://openreview.net/forum?id=pC3WJHf51j>.
- [38] Yu H, Guo P, Sano A. ECG Semantic Integrator (ESI): A Foundation ECG Model Pretrained with LLM-Enhanced Cardiological Text. Transactions on Machine Learning Research. 2024. Available from: <https://openreview.net/forum?id=giEbg8Khcf>.
- [39] Liang Y, Wen H, Nie Y, Jiang Y, Jin M, Song D, et al. Foundation models for time series analysis: A tutorial and survey. In: Proceedings of the 30th ACM SIGKDD conference on knowledge discovery and data mining; 2024. p. 6555-65.
- [40] Das A, Kong W, Sen R, Zhou Y. A decoder-only foundation model for time-series forecasting. In: Forty-first International Conference on Machine Learning; 2024. .
- [41] Goswami M, Szafer K, Choudhry A, Cai Y, Li S, Dubrawski A. MOMENT: a family of open time-series foundation models. In: Proceedings of the 41st International Conference on Machine Learning; 2024. p. 16115-52.

- [42] Zhou R, Lu L, Liu Z, Xiang T, Liang Z, Clifton DA, et al. Semi-supervised learning for multi-label cardiovascular diseases prediction: a multi-dataset study. *IEEE Transactions on Pattern Analysis and Machine Intelligence*. 2023;46(5):3305-20.
- [43] Zhang K, Wen Q, Zhang C, Cai R, Jin M, Liu Y, et al. Self-Supervised Learning for Time Series Analysis: Taxonomy, Progress, and Prospects. *ArXiv preprint*. 2023;abs/2306.10125. Available from: <https://arxiv.org/abs/2306.10125>.
- [44] Sohn K, Berthelot D, Carlini N, Zhang Z, Zhang H, Raffel CA, et al. Fixmatch: Simplifying semi-supervised learning with consistency and confidence. *Advances in neural information processing systems*. 2020;33:596-608.
- [45] Xiao Z, Tong H, Qu R, Xing H, Luo S, Zhu Z, et al. CapMatch: Semi-supervised contrastive transformer capsule with feature-based knowledge distillation for human activity recognition. *IEEE transactions on neural networks and learning systems*. 2023.